# AutoMix: Automatically Mixing Language Models

**Pranjal Aggarwal**$^{\diamond *}$  **Aman Madaan**$^{\clubsuit *}$  **Ankit Anand** $^{\ddagger}$  **Srividya Pranavi Potharaju** $^{\dagger}$
**Swaroop Mishra**$^{\ddagger}$  **Pei Zhou**$^{\triangle}$  **Aditya Gupta**  **Dheeraj Rajagopal**$^{\dagger}$  **Karthik Kappaganthu**$^{\dagger}$
**Yiming Yang**$^{\spadesuit}$  **Shyam Upadhyay**$^{\dagger}$  **Manaal Faruqui**$^{\dagger}$  **Mausam**$^{\diamond}$
$\spadesuit$ Carnegie Mellon University   $\clubsuit$ xAI   † Google   ‡ Google DeepMind
$\diamond$ IIT Delhi   $\triangle$ University of Southern California
automix-models@googlegroups.com

## Abstract

Large language models (LLMs) are now available from cloud API providers in various sizes and configurations. While this diversity offers a broad spectrum of choices, effectively leveraging the options to optimize computational cost and performance remains challenging. In this work, we present `AutoMix`, an approach that strategically routes queries to larger LMs, based on the approximate correctness of outputs from a smaller LM. Central to `AutoMix` are two key technical contributions. First, it has a few-shot self-verification mechanism, which estimates the reliability of its own outputs without requiring extensive training. Second, given that self-verification can be noisy, it employs a POMDP based router that can effectively select an appropriately sized model, based on answer confidence. Experiments across five language models and five challenging datasets show that `AutoMix` consistently surpasses strong baselines, reducing computational cost by over 50% for comparable performance. [1]

## 1 Introduction

The landscape of Large Language Models (LLMs) is rapidly evolving, with a wide array of models now available in various sizes, capabilities, and computational requirements [Touvron et al., 2023, OpenAI, 2023, Jiang et al., 2023a]. While larger models generally exhibit superior performance, their substantial computational costs render them unaffordable for many simpler tasks. Moreover, the vast array of available options makes it challenging for end-users to determine the optimal model configuration for their specific needs. This challenge is further compounded by the intrinsic complexity and variability of real-world tasks, ranging from simple (e.g., binary classification on separable data) to complex (e.g., code generation) and potentially unsolvable tasks (e.g., certain forms of multi-step reasoning). To address these issues and ensure that end-users can obtain the best performance within their budget constraints, the development of model-switching techniques has become increasingly important. These techniques involve dispatching queries to models of disparate sizes and capabilities, allowing for a more efficient allocation of computational resources [Liu et al., 2020, Zhou et al., 2020, Madaan and Yang, 2022, Geng et al., 2021, Schuster et al., 2022].

Contemporary model-switching strategies often rely on separate routing models trained for a fixed set of tasks [Chen et al., 2023, Ding et al., 2024]. Moreover, modern LLMs are frequently accessible only through black-box APIs, restricting direct model optimization due to the unavailability of fine-tuning capabilities and weight access. This constraint, coupled with the expectation of access to large amounts of task-specific data, creates a challenge that existing routing approaches fail to address adequately. In response, we introduce `AutoMix`, a method that enables users to *mix* models

---

*\*Equal Contribution. Work started and partly done during Aman's internship at Google.

[1]Code available at github.com/automix-llm/automix

38th Conference on Neural Information Processing Systems (NeurIPS 2024).

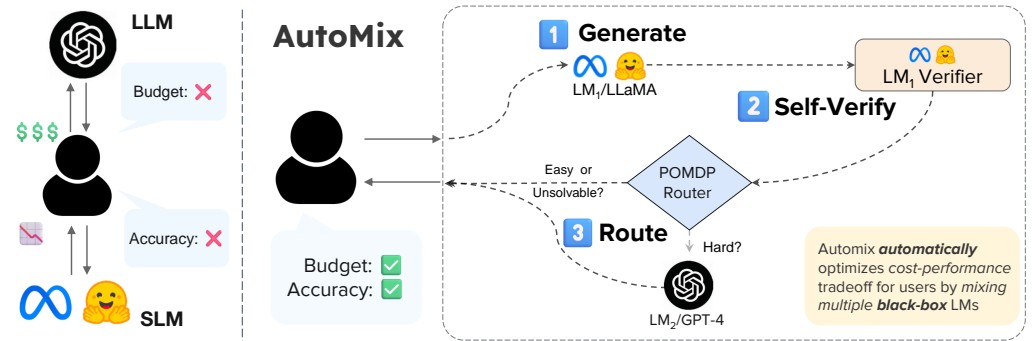

Figure 1: Representative example for 2 model setup in AutoMix. Instead of relying only on small model (SLM) with low performance or a large model (LLM) with high cost, AutoMix automatically mixes multiple black-box language models, based on user desired cost-quality tradeoff. AutoMix works in a 3-step process: 1.) generation by a small model ($LM_1$), 2.) self-verification of the generated answer, 3.) using confidence assessments from self-verification to do appropriate routing to a larger model ($LM_2$). For N-model setup, the process is repeated till the final answer is reported.

of various sizes and capabilities, assuming only access to black-box LLM APIs. As illustrated in Figure 1, AutoMix consists of 3 steps designed within the constraints of black-box access: solution generation (small model to generate initial answer), self-verification (same smaller model assesses difficulty), and selective routing (routing to larger models when suggested by self-verification). At a high level, this process mirrors human problem-solving, which inherently follows a multi-step process: generate a solution, verify its validity, and refine it further based on verification outcomes.

An ideal router in AutoMix must incorporate several key characteristics. Firstly, it should be capable of identifying the difficulty of a query based on confidence assessments of the smaller model's answers. Secondly, it should be able to route to the appropriately sized model, for instance, easy queries are routed to the small language model (SLM), hard queries to large language model (LLM), and unlike previous works [Chen et al., 2023, Ramírez et al., 2024], unsolvable queries that no model can solve should not be routed to any LM, thereby saving costs. Thirdly, it should be able to learn from a small amount of data, as expected in real-world tasks. While self-verification seems a natural choice to provide confidence assessments and learn from limited data, prior works have shown self-verification is not particularly effective for reasoning tasks and is often noisy and ill-calibrated [Tyen et al., 2023, Huang et al., 2023b] – a challenge that needs to be addressed. Finally, the router should generalize to different scenarios in terms of the number of models with varying costs and capabilities.

To address these requirements, we propose two novel contributions in AutoMix. First, we formulate self-verification as an entailment problem, where the consistency of the generated answer with the context is evaluated to estimate confidence levels [Poliak, 2020, Dagan et al., 2022]. For example, an answer discussing "apples" in a context focused on "tea" would be flagged as highly inconsistent. Second, we introduce a Partially Observable MDP (POMDP) based *router* [Åström, 1965]. POMDPs extend standard MDPs to decision problems where observations (self-verification probabilities) are unreliable and provide a noisy estimate of states (question difficulty). Modeling the router as a POMDP agent ensures all our requirements are met, as the POMDP can implicitly model various difficulties and assess them based on noisy self-verification outputs. Furthermore, POMDPs provide a principled formulation to learn robust routing policies in different scenarios with varying numbers of models, different costs, and capabilities, while learning from as few as 50 examples.

We conduct extensive evaluations of AutoMix on five different dialogue and context-grounded reasoning tasks, with five different models. Our results demonstrate that AutoMix consistently outperforms baselines while reducing cost by over two times while achieving the same performance, showcasing significant improvements and efficiency gains over existing model-switching strategies.

## 2 Background and Related Work

**Self-Verification** The concept of self-verification in reasoning problems has been explored in various works, such as Weng et al. [2023], Jiang et al. [2023b], Pan et al. [2023a]. These approaches typically

```
Context: {context}

Question: {question}

AI Generated Answer: {generated_answer}

Instruction: Your task is to evaluate if the AI Generated Answer is correct, based on the
↪  provided context and question. Provide the judgement and reasoning for each case. Choose
↪  between Correct or Incorrect.

Evaluation:"
```

Listing 1: **Verification Prompt.** The verification process is framed as a natural language entailment task, where the model determines the validity of the model-generated answer with respect to the context and question. We use a generic few-shot prompt for all tasks.

rely on the LLM's knowledge [Dhuliawala et al., 2023], a method that can pose challenges for reasoning problems [Madaan et al., 2023, Huang et al., 2023b]. In contrast, `AutoMix` leverages context for verification and introduces a POMDP router to mitigate potential noise from the verifier. Another line of work collects a corpus of past mistakes made by models [Madaan et al., 2022], and external knowledge bases for verification [Peng et al., 2023, Gao et al., 2023, Pan et al., 2023b]. In contrast, `AutoMix` uses the question's context to verify the answer. While previous works find self-verification is unreliable to repair the model's output [Huang et al., 2023a], we demonstrate that self-verification provides a valuable signal used for routing to an appropriate model.

**Mixing Models** Several works have sought to optimize LLM inference cost through model switching, employing specifically trained verifiers [Chen et al., 2023, Zhu et al., 2023, vSakota et al., 2023, Ding et al., 2024]. `AutoMix` eliminates the need for expensive verifier training through few-shot SLM model prompting and does not require upfront access to all input queries. The router, trained with as few as 50 samples, outperforms specialized models. Our work is thus aligned with recent work that aims at composing different models and external tools for inference time improvement of language models [Khattab et al., 2023, Press et al., 2022, Yao et al., 2022, Zhou et al., 2022].

**Adaptive Computation** Adaptive computation and model routing methods often preempt computation via intermediate representations [Liu et al., 2020, Zhou et al., 2020, Geng et al., 2021, Schuster et al., 2022, Madaan and Yang, 2022]. Unlike these methods, `AutoMix` requires no architectural changes and assumes only black-box access to APIs. While some black-box methods like Adaptive-Consistency [Aggarwal et al., 2023] aim to optimize inference for reasoning tasks, they are limited to a single LLM model, whereas `AutoMix` flexibly optimizes between two or more models.

**Background on Partially Observable Markov Decision Processes** POMDPs extend Markov Decision Processes (MDPs) to scenarios where an agent's state observation is partial or noisy. Defined as a tuple $\langle S, A, T, R, \Omega, O \rangle$, $S$ denotes states, $A$ actions, and $\Omega$ possible observations. The transition function $T$ gives transition probabilities between states given an action, while the observation function $P$ connects actions and states to observations. The reward function $R$ assigns rewards to actions in specific states. Agents in POMDPs maintain a belief ($b$), a probability distribution over states. This belief updates based on actions and observations. The objective in a POMDP is to find a policy $\pi : b \mapsto a \in A$ that maximizes the expected cumulative long-term reward. POMDPs have been used in various domains, including robotics, automated navigation, crowdsourcing workflows, and strategic planning [Kaelbling et al., 1998, Schwarting et al., 2018, Dai et al., 2010, Meuleau et al., 2013]. However, `AutoMix` uses POMDPs in a novel way to route queries between LMs. Our noisy self-verification outputs act as observations to the POMDP, which assesses question difficulty and routes to the appropriate LM to maximize a reward function of cost and performance.

## 3 Problem Formulation

We address the problem of selecting the appropriate language model (LM) to maximize a user-defined cost-quality tradeoff. We assume access to $N$ distinct language models: $LM_1, LM_2 \ldots LM_N$, numbered in increasing order of number of parameters. Each model has an associated cost of $C_i$ and (unknown) performance of $P_i$ for each input query. Our objective is to maximize the total performance for any total cost while dynamically invoking any $LM$ as appropriate for a given test

**Context**

Which country grows the most tea? The answer is India. It grows three times as much as China. Which country drinks the most tea? It's neither China nor Japan. It's Great Britain. [...] One customer put the bag into a pot. Then he just poured hot water over it. And the tea bag was born. **Shen Nong was the first to drink tea. (Shen was a Chinese emperor.) This was about 2737 B.C.** Shen had bad digestion. So he drank several cups of hot water daily[...] Tea became the drink of China.

**Question**

When did Shen Nong drink tea?

**Generated Answer (by LLAMA2-13B)**

He took it in 1990.

**Verifier Output (by LLAMA2-13B)**

The context does not mention that he took tea in 1990. The AI-generated answer is **Incorrect.**

Figure 2: **Context-Grounded Self-Verification using LLAMA2-13B in Action.** The example showcases the verifier, utilizing the *same model* as the answer generator, identifying and rejecting an inaccurate answer—*He took it in 1990*—by effectively leveraging the context.

point. In experiments, we compare cost-quality curves of various methods and also evaluate them using a newly proposed IBC metric (See Section 5.1).

To test our approach, we consider context-grounded reasoning tasks, such as comprehension-based QA and a variety of dialogue reasoning tasks, where given a context $\mathcal{C}$ (e.g., stories, newswire, or dialogue history) and a question $q$, the model must generate an accurate and coherent answer consistent with the provided context. Our choice of tasks is motivated by two key considerations: (1) longer queries are more computationally demanding, underscoring the need for an approach like `AutoMix` to navigate the cost-accuracy tradeoff, and (2) the context allows for cross-checking preliminary answers with available information using self-verification (described shortly). We assume only black-box access is available to the LM APIs. For training any router models, we assume access to a small training/validation dataset $\mathcal{D}_{train}$ consisting of $< \mathcal{C}, q, \hat{y}, y_{LM_i} >$ triplets, where $\hat{y}$ is the ground truth answer and $y_{LM_i}$ is the answer generated by $LM_i$.

## 4 AutoMix

At a high level, `AutoMix` consists of three steps: solution generation – using a small LM, say $LM_i$ (initially $i = 1$), to generate an answer $\mathcal{A}_s$; self-verification – using the same $LM_i$ to assess $\mathcal{A}_s$; and selective routing – picking a larger LM, $LM_j(j > i)$, when suggested by self-verification, else returning $\mathcal{A}_s$ as final answer. Figure 1 shows a representative example of the process for the two-LM case. Next, we discuss our two technical contributions in detail.

### 4.1 Self-Verification

To assess the trustworthiness of $\mathcal{A}_s$, `AutoMix` employs a few-shot verifier, $\mathcal{V}$, which validates $LM_i$'s output. Unlike existing works that perform verification by creating a new question [Weng et al., 2022, Jiang et al., 2023b], we frame verification as an entailment task [Dagan et al., 2005, Poliak, 2020, Dagan et al., 2022], to determine whether the answer generated by $LM_i$ aligns with the provided context. Specifically, the verifier gauges $v = \text{p}(\text{correct} = 1 \mid \mathcal{A}_s, \mathcal{C}, q)$, with correct = 1 indicating that $\mathcal{A}_s$ is correct. To estimate the probability, we sample $k > 1$ times from the verifier ($LM_i$) with high sampling temperature, and probability is then computed as $\frac{1}{k} \sum_{i=1}^{k} \mathbf{1}\{\text{correct} = 1\}$. We use the same 4-shot verification prompt for all tasks and do not train a verifier. Figure 1, 2 shows the prompt and an example of self-verification in action. We refer readers to Appendix B for all prompts used.

## 4.2 Router

Routing follows solution generation and self-verification. The router decides whether the output from $LM_i$ should be accepted or the query be routed to some $LM_j$ $(j > i)$ for improved performance. The router can also be interpreted as a *meta-verifier*, providing an extra layer of confidence assessment on the few-shot verifier's evaluations. Specifically, $\mathcal{V}$ ascertains whether $LM_i$'s answer is entailed by the context, making its decision without considering the inherent difficulty of the problem. For example, when handling *Unsolvable* queries, invoking a larger LM will be resource-inefficient and would not enhance performance. A good router can address this by not routing such a query further, and this decision needs to be made using the verification probability and trends from the training data.

Addressing the notable challenges of self-correction in large language models [Madaan et al., 2023, Huang et al., 2023b], `AutoMix` employs a non-LLM setup for routing and avoids escalating issues such as hallucination and reasoning errors [Dziri et al., 2023]. The router can, in principle, adopt various learning strategies, including supervised learning, reinforcement learning, and symbolic reasoning. Subsequent sections provide details of two different routing strategies in `AutoMix`.

**Thresholding**  In this simplistic routing approach for the two-model case ($N = 2$), the decision to route to $LM_2$ is based on the probability $v$ of the $LM_1$ verifier and a threshold $t$. If $v \geq t$, it returns $LM_1$'s answer, else routes the query to $LM_2$. Intuitively, a high probability indicates the verifier is confident in its decision and can be trusted. Varying $t$ can help explore cost-performance tradeoffs.

**POMDP-based Router**  A router should direct a query to larger LMs only when the performance gap justifies the cost-quality tradeoff. Given the inherent uncertainty in the true state of system performance, which remains unobserved, we formulate the router as a Partially Observable Markov Decision Process (POMDP) [Åström, 1965]. POMDPs are particularly well-suited for scenarios where observations, such as self-verification probabilities, may not be entirely reliable.

Recall that (Section 2) a POMDP is characterized by $(S, A, T, R, \Omega, O)$. In our application, the states $S$ represent the current selected $LM_i$ and performance metrics (e.g., accuracy or F-score) of various LMs on a data point, denoted as $S = \langle i, Perf_{LM_1}, Perf_{LM_2}, \ldots, Perf_{LM_N} \rangle$. The actions include either retaining the current LM's ($LM_i$) answer or routing to one of the larger LMs. Observations $\Omega$, in the form of the verifier, outputs $v$ from $LM_i$, enabling the POMDP to ascertain its belief state $b$: a probability distribution over $S$. The observation probabilities $P(o|s)$, which indicate the likelihood of observing $o$ (verification output) given state $s$, are crucial for defining the POMDP model. E.g., high verifier confidence might suggest that the current LM's performance $Perf_{LM_i}$ is high enough, reducing the necessity to switch to a more costly LM. The observation probabilities are estimated directly on the train set, by computing the expectation of verification probabilities for each state, i.e $P(o|s) = \frac{\sum_{s_j, v_j \in D_{train}} \mathbf{1}\{s_j = s \text{ and } v_j = o\}}{\sum_{s_j \in D_{train}} \mathbf{1}\{s_j = s\}}$. However, since the states can be continuous, we use non-parametric Gaussian kernel density estimation to estimate the observation probability for a new state. Instead of directly estimating $P(o|s)$, we first learn a joint distribution $P(S, O)$ and $P(S)$, by drawing KDE at each training point. Then conditional probability $P(o|s)$ is computed as: $P(o|s) = \frac{P(s,o)}{P(s)}$.

The objective of our POMDP is to optimize a reward function $R = P - \lambda \cdot C$, where $P$ is overall performance, $C$ is overall cost, $\lambda$ is a tuning parameter that balances two criteria, according to user preferences. We utilize the AdaOps POMDP solver [Wu et al., 2021], which employs particle filtering for representing belief, where each particle corresponds to a particular state. At inference, the POMDP solver starts with an initial belief (uniform distribution), updates its belief state based on verifier observations, and, based on the updated belief state $b'$, takes an action to maximize the expected reward. We provide further details and the complete POMDP formulation in Appendix A.3.

## 5 Experiments

Through our experiments, we wish to answer the following research questions. How does `AutoMix` compare with other model-switching strategies? How well does `AutoMix` perform when varying (1) cost ratios between models, and (2) the amount of available training data? Following recent work [Ding et al., 2024], we perform most experiments in the setting when $N = 2$, but also report

```
procedure ANSWERQUERY(C, q)
            ▷ C: Context, q: Question, SLM/LLM:
Small/large language model
    A_s ← SOLVE(SLM, C, q)
    v ← SELF-VERIFY(A_s, C, q)
    if META-VERIFY(v, A_s, C, q) then
        return A_s
    else
        A_l ← SOLVE(LLM, C, q)
        return A_l
    end if
end procedure
```

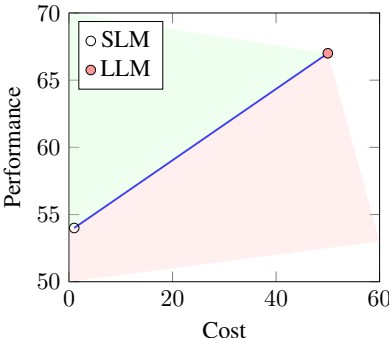

Figure 3: **Left:** AutoMix algorithm. **Right:** Performance vs. Cost curve. The slope between SLM and LLM provides a way to the Incremental Benefit per Cost (IBC) for methods that mix models. Methods with a steeper slope than this reference when plotted against SLM have a positive IBC (green region), whereas those below the reference have a negative IBC (red region).

additional results in the three model cases (in Section 5.5). For $N = 2$, we use the terms SLM and LLM to denote small ($LM_1$) and large ($LM_2$) language models, respectively.

## 5.1 A Metric for Cost-Performance Efficiency Analysis

**Incremental Benefit Per Cost (IBC)**   In our approach to leveraging model performance, it is essential to consider not only the raw accuracy of predictions but also the associated computational or monetary costs. To that end, we introduce a metric to understand the performance of the models in terms of cost. We introduce methods, denoted by $M$, to optimally integrate SLM and LLM. For each method $M$, we associate a cost $C_M$ and performance $P_M$. To quantify the utility of $M$ over SLM, we define the metric *Incremental Benefit Per Cost* (IBC) as $\text{IBC}_M$ (Equation (1)).

$$\text{IBC}_M = \frac{P_M - P_\text{SLM}}{C_M - C_\text{SLM}}, \quad \text{IBC}_\text{BASE} = \frac{P_\text{LLM} - P_\text{SLM}}{C_\text{LLM} - C_\text{SLM}}, \quad \Delta_\text{IBC}(M) = \frac{\text{IBC}_M - \text{IBC}_\text{BASE}}{\text{IBC}_\text{BASE}} \times 100 \tag{1}$$

The IBC metric captures the efficiency of performance enhancement relative to the additional cost. For comparative evaluation, we set a baseline IBC, $\text{IBC}_\text{BASE}$, representing the benefit of *always* using LLM over SLM. Finally, we compare methods using $\Delta_\text{IBC}$, which compares the IBC of a specific method with $\text{IBC}_\text{BASE}$. A positive IBC lift suggests that $M$ achieves performance increments more cost-effectively than a standalone LLM, whereas a negative lift indicates reduced efficiency (Figure 3)

**Geometric Interpretation**   On a Performance vs. Cost plot, consider the line segment joining the data points of the small language model (SLM) and the large language model (LLM). This segment's slope represents the rate of performance increase per unit of cost. The Incremental Benefit per Cost (IBC) for any method $M$ is the slope of the line from the SLM point to the point representing $M$(Figure 3). A method $M$ that lies above the SLM-LLM segment provides a steeper slope, indicating a favorable IBC (and a positive $\Delta_\text{IBC}$). Conversely, if $M$ lies below the segment, it suggests an unfavorable or negative IBC. Our primary objective is to develop methods that yield a consistently positive IBC, maximizing performance enhancements for each additional unit of cost.

## 5.2 Setup

**Models and Cost Calculation**   We use GPT-3.5, LLAMA2-13B, and MISTRAL-7B (instruct v0.2 version) [Touvron et al., 2023, Jiang et al., 2023a] as the smaller language models (SLMs), and GPT-4 [OpenAI, 2023] as the larger language model (LLM). For each input query, we model fixed costs for the models and verification, denoted by $C_\text{SLM}$ for smaller models, $C_\text{LLM}$ for the larger model, and $C_{ver}$ for the verification model. As the verification is performed by $C_\text{SLM}$ and the major cost driver is the question context, which remains the same in both models, we set $C_{ver} = C_\text{SLM}$. At

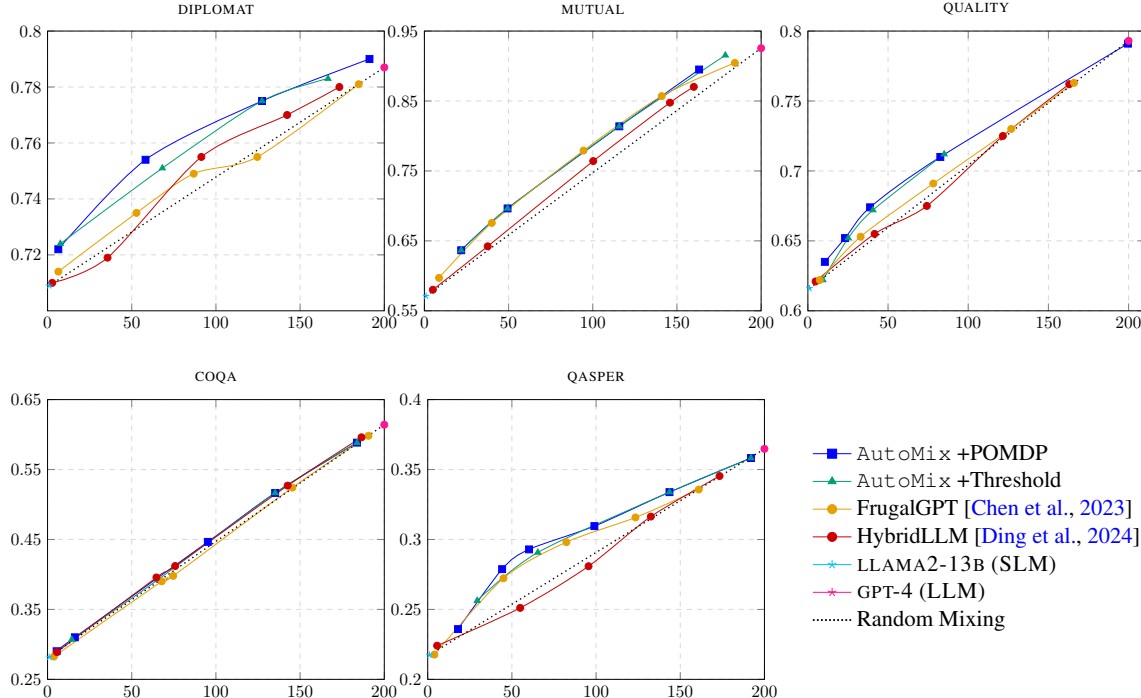

Figure 4: **Main Results:** performance **(y-axis)** vs. cost **(x-axis)** for different methods on the small and large LLAMA2-13/GPT-4. POMDP based meta-verifier is consistently above the linear interpolation (random mixing) of SLM-LLM, signifying a higher incremental benefit per unit cost (IBC).

inference, the total cost is computed as the sum of individual costs, $C = C_{\text{SLM}} + C_{ver} + w \cdot C_{\text{LLM}}$, where $w$ indicates whether the larger model is invoked.

While the pricing of these APIs [Dehghani et al., 2021] is influenced by various complexities, our focus on the black-box utilization of language models leads us to represent cost based on the actual API price disparity between these models.[2] For each of GPT-3.5, LLAMA2-13B, MISTRAL-7B, we assign a relative cost of 1 unit for the SLM and 60, 100, 200 units for the LLM respectively. It's important to note that the cost ratio between models can vary significantly depending on specific deployment scenarios. For example, for a user with access to a single A6000 GPU, running LLAMA2-13B might incur virtually no cost, while utilizing GPT-4 could be prohibitively expensive. We simulate this scenario in Section 5.4. We refer readers to Appendix B for additional details.

**Datasets** We experiment with a diverse set of datasets: i) QASPER [Dasigi et al., 2021]: Question answering over research papers; ii) QUALITY [Pang et al., 2022]: Multiple-choice questions (MCQ) on long articles and stories; iii) COQA [Reddy et al., 2019]: Conversational comprehension requiring coreference and pragmatic reasoning; iv) MUTUAL [Cui et al., 2020]: Multi-turn dialogue reasoning (next response prediction); v) DIPLOMAT [Li et al., 2023]: Pragmatic identification and reasoning questions on multi-turn dialogues. We use the F1 score for QASPER and COQA, and accuracy for the remaining datasets. We use the default validation splits and utilize prompts from Shaham et al. [2023] for QASPER and QUALITY, and adapt the QUALITY prompt for other datasets. We use identical prompts for all models. We refer readers to Appendix B for more details.

**Baselines** We compare against FrugalGPT [Chen et al., 2023] and HybridLLM [Ding et al., 2024], two state-of-the-art models, as our baselines. FrugalGPT uses a finetuned DistillBert model [Sanh et al., 2019] as a router. If the router's confidence for a given question, context, and SLM answer falls below a threshold, the query is routed to the LLM. HybridLLM uses trained DeBERTa [He et al., 2021] as a router which directly chooses between SLM and LLM without

---

[2]https://openai.com/pricing, https://together.ai/

| Model | Method | DIPLOMAT | MUTUAL | COQA | QASPER | QUALITY |
|-------|--------|----------|--------|------|--------|---------|
| MISTRAL-7B | FrugalGPT | 16.8 | 20.3 | -16.7 | 7.7 | 8.1 |
| | HybridLLM | 67.1 | 3.4 | 10.5 | 0.4 | 2.7 |
| | AutoMix + T | 149.7 | **46.7** | 12.1 | 67.9 | 33.6 |
| | AutoMix + P | **156.8** | **46.7** | **12.4** | **69.3** | **51.6** |
| LLAMA2-13B | FrugalGPT | -7.0 | -8.7 | 2.6 | **9.4** | -4.7 |
| | HybridLLM | 3.8 | 2.2 | -0.5 | 7.2 | 6.5 |
| | AutoMix + T | 50.1 | 11.8 | **86.5** | -0.2 | 9.4 |
| | AutoMix + P | **58.5** | **12.4** | 83.1 | 8.5 | **10.3** |
| GPT-3.5 | FrugalGPT | 30.1 | 11.1 | 37.0 | 87.2 | 10.1 |
| | HybridLLM | 8.3 | 1.8 | 20.1 | 21.7 | 11.8 |
| | AutoMix + T | **168.2** | 18.3 | 62.7 | 109.8 | 23.5 |
| | AutoMix + P | 151.2 | **18.8** | **65.0** | **114.7** | **24.1** |

Table 1: $\Delta_{\text{IBC}}$ **values:** AutoMix + T and AutoMix + P are variations of our proposed method with thresholding (T) and POMDP-based routers, respectively. Best numbers are bold, and second-best are underlined. **AutoMix + POMDP** demonstrates robust and consistent $\Delta_{\text{IBC}}$ across all datasets, implying judicious utilization of computational resources. Despite domain-specific training and a 0-cost verifier, FrugalGPT and HybridLLM underperform AutoMix in almost all scenarios.

generating an SLM response. Further, training labels for HybridLLM's router are generated based on the probability of SLM outperforming LLM by a margin, computed by drawing multiple samples. Note, that training each of these baselines along with AutoMix requires running inference on all models, however, it is only a training time cost. However, at inference, unlike AutoMix we assign a cost of 0 to the two baselines' routers due to their lower operational costs.

## 5.3 Main Results

Figure 4 illustrates the performance versus cost curves for various datasets and model-mixing methods using MISTRAL-7B as SLM. Across all datasets, AutoMix-POMDP and AutoMix-Threshold consistently outperform FrugalGPT and HybridLLM, staying above the SLM-LLM curve and yielding better performance per unit cost. This is particularly notable as both FrugalGPT and HybridLLM leverage domain-specific trained routers and do not incur any verification costs. Comparable trends are observed with other SLMs LLAMA2-13B and GPT-3.5 in Figure 18, 19.

Table 1 presents a comparison of the average $\Delta_{\text{IBC}}$ across five equally sized cost regions for each method, dataset, and three different SLMs. AutoMix-POMDP consistently outperforms FrugalGPT and HybridLLM across all datasets and models, with performance equalling FrugalGPT in the qasper dataset for the LLAMA2-13B model. The gains vary across different settings, with the best performance observed on the DIPLOMAT dataset, averaging 122% across all models. Importantly, AutoMix-POMDP is the only method that yields positive gains across all configurations and consistently matches or exceeds the performance of AutoMix-Threshold. This is despite being tested on a variety of datasets and models representing different costs, task difficulty (accuracy ranging from 30% to 90%), performance gap between the models (8% to $\approx 50\%$), different answer types (MCQ, Open-ended). The results demonstrate that AutoMix, utilizing self-verification and appropriate routing, can effectively mix different SLMs and GPT-4 on a wide range of tasks without access to model weights or domain-specific routing data. These substantial gains translate directly into cost savings, underscoring the economic relevance of our approach.

## 5.4 Additional Experiments

**Effect of Cost Ratio on AutoMix** Our main experiments assumed different cost ratio ranging from 1:60 to 1:200 for different SLMs. Subsequently, we conduct an analysis to understand how alterations in the cost ratio influence the $\Delta_{\text{IBC}}$ values across different cost ratios. In Figure 5 (right), we vary cost-ratio and report $\Delta_{\text{IBC}}$ normalized by the maximum value obtained on the dataset (a higher value denotes higher gains). The results suggest that even for a cost ratio as low as 1:10, AutoMix begins to deliver good gains for many datasets, which naturally improve as cost-ratio gets further skewed.

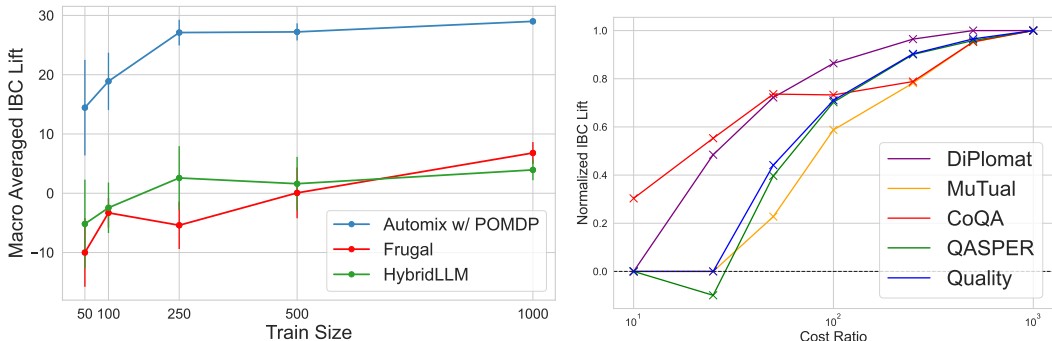

Figure 5: **Left:** Comparison of `AutoMix` with `FrugalGPT` and `HybridLLM` over varying training data sizes shows that despite 0-cost verifier and domain-specific training, baselines underperform `AutoMix`. `AutoMix` excels in limited data settings. **Right:** Normalized $\Delta_{\text{IBC}}$ across different cost ratios. `AutoMix` show robust performance gains even when cost ratio is low.

We supplement the analysis with cost-performance curves for different cost ratios in Appendix D.2. We note that cost need not only be monetary cost, and could, in general, represent other considerations such as latency or energy usage. These results demonstrate that `AutoMix` provides a novel method to balance the cost-performance tradeoff across a wide range of cost scenarios and is robust to changes without any modifications to the method.

**AutoMix is Effective in Low-Resource Scenarios**    Figure 5 (left) demonstrates the performance dynamics of `AutoMix`, `FrugalGPT` and `HybridLLM` with varying validation set size. Notably, our method significantly outperforms `FrugalGPT` and `HybridLLM` with limited data, despite the latter's domain-specific training and zero verifier cost. E.g., with just 50 training examples, `AutoMix` maintains 15% $\Delta_{\text{IBC}}$ and beats the baselines by more than absolute 20%. This underscores that `AutoMix` is particularly advantageous in the real-world scenarios where training data is scarce.

**When Does POMDP Routing Help?**    POMDP implicitly categorizes question difficulty into three types: (a) *easy* queries solvable by the small model, (b) *complex* queries only larger models can solve, and (c) *potentially unsolvable* queries for any model. It leverages self-verification probabilities—which provide a noisy estimate of difficulty—to make non-linear decisions. For example, in the QASPER dataset using MISTRAL-7B as SLM, POMDP identifies that lower confidence values correspond to such cases, and instead of routing to the LLM as other methods do, returns the SLM answer, saving cost. Furthermore, in setups with more than two models (as discussed in Section 5.5), POMDP makes more nuanced decisions, for instance by combining information from small and medium-sized verifiers, resulting in significantly superior performance.

**Other Experiments**    We perform various other experiments, whose details are in appendix due to space limitations. We first evaluate few-shot self-verification quantitatively and qualitatively and observe that the self-verification can effectively use context to identify errors in answers generated by SLM in many cases (see Appendix A).

While `AutoMix` is outperforms on diverse set of datasets, to further demonstrate our router's generalizability, we evaluate out-of-domain generalization by training on one dataset and evaluating on held-out datasets. Results in Appendix D.3 show that `AutoMix` consistently outperforms `FrugalGPT` and `HybridLLM` by significant margins. Overall, the task-agnostic design of POMDP and these results highlight our method's generalizability.

We also study the latency overhead of `AutoMix`. At inference time, POMDP takes less than 1ms for every query in the two-model setup. Moreover, network latency (usually approximately 10ms) is much less than the average solution generation time by SLM and LLM (on the order of seconds). Thus, `AutoMix` adds no additional computational or latency overhead compared to language model inference. Finally, incorporating the POMDP router for end-users is convenient, as our library can be incorporated in less than five lines of code.

## 5.5 Results of Automix with Three Models

We evaluate the performance of `AutoMix` applied to a three-model scenario ($N = 3$). Specifically, we use LLAMA2-13B as SLM, LLAMA2-70B as MLM, and GPT-4 as LLM. Results are presented in Figure 6. `AutoMix` consistently outperforms baselines across cost regions. We first compare `AutoMix` to `FrugalGPT`, showing significant improvements across all cost regions considered. We also compare `AutoMix` to a baseline, *Union* `AutoMix`, which selects between `AutoMix`_{SLM−MLM} and `AutoMix`_{MLM−LLM} based on the user's cost requirements. For instance, if the desired average cost is less than that of the MLM, `AutoMix`_{SLM−MLM} is employed; otherwise, `AutoMix`_{MLM−LLM} is utilized. Further, we consider a baseline, Chained `AutoMix`, by chaining `AutoMix`_{SLM−MLM} with `AutoMix`_{MLM−LLM}. The query first goes to the SLM, and an `AutoMix`_{SLM−MLM} decides between reporting the SLM answer or routing to the MLM. In the latter case, a second

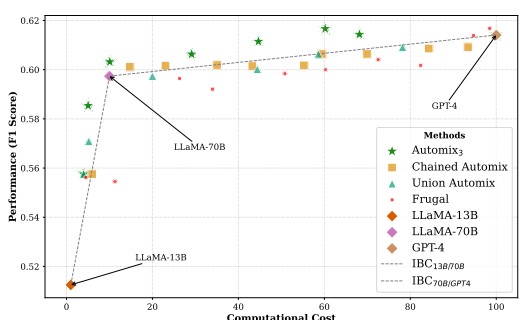

Figure 6: `AutoMix` with 3 models: LLAMA2-13B, GPT-4 and GPT-4. `AutoMix` method shows consistent IBC lifts for both SLM-MLM and MLM-LLM regions. Further, compared to baselines: `FrugalGPT`, chaining two `AutoMix` models or using the union of two `AutoMix`es, `AutoMix`_3 provide significant improvements.

`AutoMix`_{MLM−LLM} repeats the procedure using the MLM and LLM models. Chained `AutoMix` underperforms across the board, as it cannot directly route queries from the SLM to the LLM. Additionally, whenever Chained `AutoMix` prompts the MLM, it invariably uses the costly verifier, even when it might not be necessary. Further, in Figure 9, 10, we study two additional cases: 1.) If the MLM underperforms the SLM, and its verifier is uninformative, then unlike `FrugalGPT`, `AutoMix` learns the irrelevance of MLM and routes directly from SLM to LLM as needed; 2.) In scenarios where MLM underperforms the SLM, but its verifier provides useful information, `AutoMix` leverages information from MLM's verifier to outperform all baselines considered. The experiments show the efficacy of `AutoMix` in providing optimum performance across diverse situations without additional intervention. We refer readers to Appendix C for more details.

## 6 Conclusion

`AutoMix` integrates multiple black-box LLM APIs into a multi-step problem-solving framework, optimizing the computational cost and performance trade-offs. Our work interweaves Good Old-Fashioned Artificial Intelligence (GOFAI) approaches with LLMs, demonstrating that the incorporation of a POMDP can lead to effective routing between LLMs. Our work provides a novel method to balance cost-performance tradeoffs across a wide variety of models over different desired cost ranges, consistently outperforming the baselines by significant margins. `AutoMix` opens avenues for several interesting research directions. While self-verification and correction are challenging for LLMs in general, we find promising results using context-grounded few-shot verification coupled with POMDPs, indicating that similar approaches may also help other scenarios. The work may be extended to generation tasks, where the notion of performance is subjective.

## Limitations

While our empirical evidence across various models and datasets demonstrates the effectiveness of `AutoMix`, its broader applicability might vary depending on the specific models and datasets used. Furthermore, `AutoMix` is designed with a dialogue-related or context-grounded reasoning setup in mind for effective self-verification, which does not include tasks like factual question-answering and commonsense reasoning. In the future, as open-source models become more powerful and inference costs decrease, it might be feasible to serve a strong model for all queries. However, there are likely to still be availability trade-offs that could be managed using `AutoMix`.

# 7 Acknowledgements

This work is supported by the IBM AI Horizons Network grant, grants by Google, and Microsoft, an IBM SUR award, and the Jai Gupta chair fellowship by IIT Delhi. We thank the IIT Delhi HPC facility for its computational resources. We are grateful to Microsoft AFMR for supporting this work. We also thank Kalpesh Krishna, Prakhar Gupta, Rahul Gupta, Siddharth Gopal, and Yang Song for their valuable feedback.

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

# A How effective is few-shot self-verification?

## A.1 Quantitative Analysis

One notable contribution of this work is the concept of few-shot self-verification of outputs. Self-Verification, especially for reasoning problems, poses its own set of challenges; however, our setup has a unique advantage: the capacity to utilize context to validate answers. For instance, the model can identify factual inaccuracies in the answer or discern apparent contradictions that might not have been evident during the initial response. But does this advantage translate to effective self-verification in practice? As depicted in Figure 12, for all but once few-shot self-verification succeeds in accurately identifying correct examples by assigning them higher probabilities across all other datasets. Specifically, the y-axis on the graph denotes the precision of identifying the correct responses (defined as F1 > 0.1). For all but QUALITY and QASPER higher verification probability, always higher precision. Even for QASPER, barring 2/9 outliers, the trend generally holds.

## A.2 Qualitative Analysis

Representative Examples from our qualitative analysis are shown in Tables 2, 3, and 4.

| Attribute | Content |
|---|---|
| Context | Which country grows the most tea? ... Shen sipped it. He liked it. He drank it all. Shen was proud of his new drink. *Truncated for brevity* |
| Question | When did he take it? |
| Generated Answer | He took it in 1990. |
| Gold Answer | about 2737 B.C |
| Verifier Output | The context does not mention that he took tea in 1990. Verification Decision: The AI generated answer is Incorrect. |

Table 2: An example where the generated answer is verifiably incorrect, citing an unsupported date (1990), and the verifier successfully catches this discrepancy.

| Attribute | Content |
|---|---|
| Context | Fort Lauderdale, Florida (CNN) – Just taking a sip of water or walking to the bathroom is excruciatingly painful for 15-year-old Michael Brewer, ... He suffered second- and third-degree burns over about two-thirds of his body, according to the hospital's associate director, Dr. Carl Schulman. *Truncated for brevity* |
| Question | How long before he recovers from his injury? |
| Generated Answer | he will never recover from his injury. |
| Gold Answer | He has a lifelong recovery. |
| Verifier Output | The context clearly states that the teen is recovering from his injuries. Verification Decision: The AI generated answer is Incorrect. |

Table 3: The example demonstrates a challenging case for the verifier. The generated answer is a rephrasing of what's mentioned in the context, however the verifier fails to catch it.

**FrugalGPT vs. `AutoMix` at different levels of data availability**  Figure 7 demonstrates the performance dynamics of `AutoMix` and FrugalGPT with varying validation data sizes. Notably, our method significantly outperforms FrugalGPT with limited data, despite the latter's domain-specific training and zero verifier cost. However, as training data increases, FrugalGPT narrows the

| Attribute | Content |
|---|---|
| Context | Pruning keeps the plant only three or four feet tall. This is an easy height for tea picking. Green tea is steamed right after the leaves are picked. Green tea _ its healthy qualities. For example, it may prevent heart disease. How did we get tea bag? The answer: by accident. *Truncated* |
| Question | What did they do to green tea after picking it? |
| SLM Answer | steamed. |
| Gold Answer | prune it |
| Verifier Output | The context mentions that green tea is steamed after being picked. Verification Decision: The AI generated answer is Correct. |

Table 4: An instance where the verifier deems the answer correct, and the gold label was incorrect. The verifier is able to correctly infer that the context mentions tea being steamed after picking.

performance gap by leveraging its domain-specific training. This pattern indicates that `AutoMix` provides a particularly advantageous solution in real-world scenarios where data may be scarce.

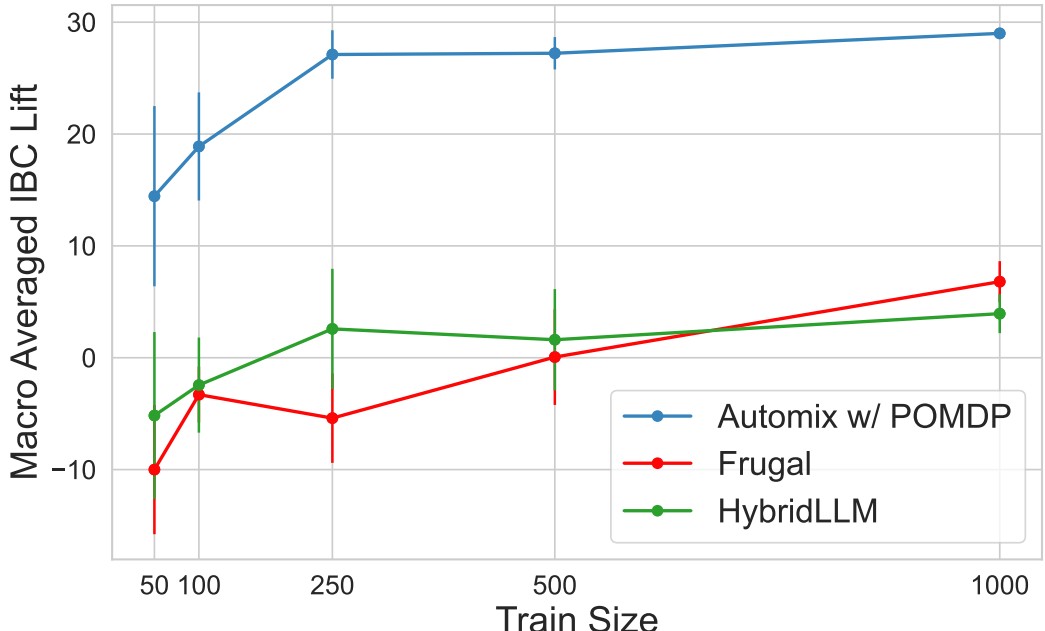

Figure 7: Comparison of `AutoMix` with FrugalGPT over varying Training Dataset Size. Despite zero-cost verifier and domain-specific training, `FrugalGPT` underperforms `AutoMix`. `AutoMix` is especially useful for limited data settings, with higher gains visible when dataset size is less than 1000.

### A.3 POMDP-based Router in `AutoMix`

A router should direct a query to larger Language Models (LMs) only when the performance gap justifies the cost-quality tradeoff. Given the inherent uncertainty in the true state of system performance, which remains unobserved, we formulate the router as a Partially Observable Markov Decision Process (POMDP) [Åström, 1965]. POMDPs are particularly well-suited for scenarios where observations, such as self-verification probabilities, may not be entirely reliable.

Recall that (Section 2) a POMDP is characterized by the tuple $(S, A, T, R, \Omega, O)$. In our application, we define these components as follows:

**States** ($S$): The state space represents the current selected $LM_i$ (stage) and performance metrics (e.g., accuracy or F-score) of various LMs on a data point. We denote this as $S = \langle i, Perf_{LM_1}, Perf_{LM_2}, \dots, Perf_{LM_N} \rangle$, where $i$ is the index of the current LM and $Perf_{LM_j}$ is the performance of the $j$-th LM. The terminal state is represented by $S = \langle N, \dots \rangle$, where $N$ is the total number of LMs.

**Actions** ($A$): The action space includes either retaining the current LM's answer ($a = LM_i$) or routing to one of the larger LMs (denoted by $Route_j, j > i$). For all LMs except $LM_N$, routing to an $LM_j$ implies performing both answer generation and self-verification.

**Transition Function** ($T$): The transition probabilities are deterministic since the true performance of language models on a question is static, albeit unknown. However, the first dimension in the state vector, representing the current $LM_i$, changes to $j$ if action $Route_j$ is taken or to $N$ if the current LM's answer is retained. As the stage state always increases and is bounded by $N$, our problem has a finite horizon.

**Reward Function** ($R$): The objective of our POMDP is to optimize a reward function $R = P - \lambda \cdot C$, where $P$ is overall performance, $C$ is overall cost, and $\lambda$ is a tuning parameter that balances these two criteria according to user preferences.

**Observations** ($\Omega$): Observations come in the form of verifier outputs $v$ from $LM_i$, enabling the POMDP to ascertain its belief state $b$, which is a probability distribution over $S$.

**Observation Function** ($O$): The observation probabilities $P(o|s)$ indicate the likelihood of observing $o$ (verification output) given state $s$. These probabilities are crucial for defining the POMDP model. For instance, high verifier confidence might suggest that the current LM's performance $Perf_{LM_i}$ is sufficiently high, reducing the necessity to switch to a more costly LM.

To estimate the observation probabilities, we first consider the discrete case. In this scenario, we can directly compute $P(o|s)$ using:

$$P(o|s) = \frac{\sum_{(s_j, v_j) \in D} \mathbf{1}\{s_j = s \text{ and } v_j = o\}}{\sum_{s_j \in D} \mathbf{1}\{s_j = s\}}$$

where $D$ represents the set of state-observation pairs from the training data.

However, in our continuous state space, we need to adapt this approach. We employ Kernel Density Estimation (KDE), a non-parametric method for estimating probability density functions. Instead of directly estimating $P(o|s)$, we first learn a joint distribution $P(S, O)$ using KDE. For this, we draw a KDE at each training point:

$$\hat{f}_h(x) = \frac{1}{nh} \sum_{i=1}^{n} K\left(\frac{x - X_i}{h}\right)$$

where $K$ is a Gaussian kernel function, $h$ is the bandwidth, and $X_i$ are the sample points.

The conditional probability $P(o|s)$ is then computed as:

$$P(o|s) = \frac{P(s, o)}{P(s)}$$

where $P(s)$ is computed similar to $P(s, o)$ These observation probabilities are estimated directly on the training set by drawing KDEs for each training point.

By leveraging this POMDP formulation and the KDE-based observation model, our router can make informed decisions about when to route queries to larger language models, effectively balancing performance and cost considerations in the face of uncertainty.

### A.3.1 Solving POMDP

Once all the different components of our POMDP are defined as shown above, finding an optimal policy requires a suitable solver. An optimal policy in this context is a strategy $\pi^* : B \to A$,

where $B$ is the set of all possible belief states that maximizes the expected cumulative discounted reward. However, our formulation presents unique challenges and opportunities that traditional POMDP algorithms struggle to address efficiently. Our POMDP models a continuous state space in an $(N + 1)$-dimensional space, where $N$ is the number of language models. This continuous nature poses difficulties for many traditional POMDP algorithms, such as point-based value iteration methods, which are typically designed for discrete state spaces and can become computationally intractable in continuous domains.

Nonetheless, our particular formulation has several desirable characteristics that distinguish it from traditional POMDPs: 1. *Finite and small horizon:* Our problem has a limited number of decision stages (maximum = $N$), bounding the depth of any decision tree we need to explore. 2. *Finite and small number of observations:* The verifier outputs, which serve as our observations, are discrete and limited in number (9), allowing enumeration of all possible observation sequences. 3. *Directed Acyclic Graph (DAG) structure:* In our formulation, a particular state is encountered only once, and the problem progresses unidirectionally through the stages.

These characteristics allow us to detail a more tailored and efficient solution approach. We first start by presenting a simple yet effective algorithm to solve our POMDP, which leverages these properties to achieve decent speed and accuracy. The approach combines particle filtering for belief state representation with a tree-based search strategy, as detailed in the following section.

**POMDP Solver** We start with a description of a simple POMDP solver to understand the intuition behind how to solve POMDPs with our proposed formulation. Because of the multidimensional continuous state space, we cannot represent all possible belief states explicitly. A common method to approximate belief states in such cases is the use of particle filters.

Specifically, each particle represents a possible state of the system, and the belief state is represented as a set of such particles. Mathematically, if we denote the set of particles by $\{s^{(i)}\}_{i=1}^{M}$, where $M$ is the number of particles, the belief state $b$ at any time is approximated as

$$b(s) \approx \frac{1}{M} \sum_{i=1}^{M} \delta(s - s^{(i)})$$

where $\delta$ is the Dirac delta function.

To solve our POMDP, we construct a decision tree that represents the possible sequences of actions and observations. This tree consists of two types of nodes: action nodes, which represent the selection of a language model, and observation nodes, which represent the possible verifier outputs following an action. After each observation node, we update the belief particles using the observation probabilities, refining our estimate of the system's state.

Assuming that we can explore and expand all nodes for small trees, we need to compute the value of each node. We can compute the reward directly from the reward model for terminal nodes. For any other node, the value $V(b)$ is defined as the expected reward of the best action, which can be recursively computed using:

$$V(b) = \max_{a} \left[ R(b, a) + \gamma \sum_{o} P(o \mid b, a) V(b') \right]$$

where $b'$ is the updated belief state after observing $o$ following action $a$, and $\gamma$ is the discount factor set to 1 in our case.

One final step remains: how to update the belief states. This is done using particle filtering, as described next. In particle filtering, belief states are updated by sampling from the current belief state based on the observation probabilities.

Mathematically, unweighted particle filtering updates can be described as follows. Given a set of particles $\{s_t^{(i)}\}_{i=1}^{M}$ representing the belief state at time $t$, and an observation $o_t$, the updated set of particles $\{s_{t+1}^{(i)}\}_{i=1}^{M}$ is obtained by:

1. For each particle $s_t^{(i)}$, sample a new state $s_{t+1}^{(i)}$ according to the transition model $P(s_{t+1} \mid s_t^{(i)}, a_t)$.

2. Weight each new particle by the observation probability $P(o_t \mid s_{t+1}^{(i)})$.

3. Resample $M$ particles from the weighted set to form an unweighted particle set.

The algorithm can be summarized as follows:

---

**Algorithm 1** Unweighted Particle Filtering Update

---

1: **Input:** Current particles $\{s_t^{(i)}\}_{i=1}^M$, action $a_t$, observation $o_t$
2: **Output:** Updated particles $\{s_{t+1}^{(i)}\}_{i=1}^M$
3: **for** each particle $s_t^{(i)}$ **do**
4:     Sample $s_{t+1}^{(i)} \sim P(s_{t+1} \mid s_t^{(i)}, a_t)$
5:     Compute weight $w_i = P(o_t \mid s_{t+1}^{(i)})$
6: **end for**
7: Resample $M$ particles from $\{s_{t+1}^{(i)}, w_i\}_{i=1}^M$ to obtain unweighted particles $\{s_{t+1}^{(i)}\}_{i=1}^M$

---

Given our small and finite horizon problem, we set the discount factor as 1. Finally, we can simulate all possible scenarios and learn deterministic policies offline because of a finite number of observation sequences. Thus, POMDP is reduced to a simple lookup table at inference, and the inference cost is negligible.

**Using better POMDP solvers**

Note that the simple solver design introduced in the previous section is sufficient and fast enough for our purpose. However, to handle more complicated scenarios in future works, we use a more advanced and recently introduced AdaOps POMDP solver [Wu et al., 2021]. AdaOps, which stands for Adaptive Online Packing-guided Search for POMDPs, offers several advantages, such as efficient handling of larger state spaces and improved scalability through adaptive online search techniques. Additionally, it incorporates packing-guided exploration strategies to focus computational resources on the most promising parts of the belief space, leading to more accurate policy estimations in complex environments or where search and state space is large. However, given our simpler formulation, both AdaOps and the simple algorithm would approximately converge to the same solution as the number of particles used in belief state representation increases.

# B    Additional Details on the Experimental Setup

For evaluation, we utilize the validation sets from Shaham et al. [2022] for QASPER, and QUALITY, and use the prompts from Shaham et al. [2023]. For COQA, MUTUAL, and DIPLOMAT, we employ its validation split and adapt the QUALITY prompt. For consistency, 1000 instances are sampled from the validation set of each dataset. Regardless of the dataset, identical input prompts are dispatched to both SLM and potentially LLM, ensuring consistent input processing costs. The output length is fixed in multichoice datasets like CNLI and QUALITY, and the brevity of responses in other datasets allows us to assume uniform output processing costs. We use greedy decoding (temperature 0) and draw a single sample for both the SLM and LLM. For verification, we generate eight samples per question (temperature = 1), which has negligible cost owing to the large context. In Figure 6, we normalize $\Delta_{\text{IBC}}$ by a scaling factor such that for all datasets, the maximum is set to 1.

For running our experiments, we use LLAMA2-13B and GPT-4 models from huggingface[3]. We use vllm [Kwon et al., 2023] for hosting models for inference.

**Cost Ratio:**    We have considered a cost ratio of 1:100 between GPT-4 and GPT-4, reflecting the API price disparity between the models, which stands at $0.225 for LLAMA2-13B vs $30 for GPT-4 per 1M tokens at the time of writing. Additionally, for self-verification purposes, we generate 8 samples. It is important to note, however, that the cost of generating 8 samples is negligible compared to the cost of a single sample, primarily because the major cost driver is the length of the context

---

[3]Models available at: https://huggingface.co/meta-llama/Llama-2-13b-hf and https://huggingface.co/meta-llama/Llama-2-70b-hf

(e.g., generation is 60 times and 50 times smaller for QASPER and QUALITY, respectively than base context). Therefore, invoking the verifier 8 times is considered equivalent in cost to calling it once. Furthermore, in Section 6, we explore different cost ratios and observe that, even with a ratio as low as 1:25, AutoMix begins to yield non-trivial gains across most datasets.

**Datasets** We experiment with a diverse set of datasets: i) QASPER [Dasigi et al., 2021]: Question answering over research papers; ii) QUALITY [Pang et al., 2022]: Multiple-choice questions (MCQ) on long articles and stories; iii) COQA [Reddy et al., 2019]: Conversational comprehension requiring coreference and pragmatic reasoning; iv) MUTUAL [Cui et al., 2020]: Multi-turn dialogue reasoning (next response prediction); v) DIPLOMAT [Li et al., 2023]: Pragmatic identification and reasoning questions on multi-turn dialogues. QUALITY, QASPER and MUTUAL are licensed under CC BY, while DIPLOMAT is licensed under CC BY-NC-SA. COQA uses multiple licenses for several splits and are detailed in https://stanfordnlp.github.io/coqa/. We use the F1 score for QASPER and COQA, and accuracy for the remaining datasets. To manage input complexity, we retain a context subset (max 3500 tokens) retrieved using the question as a key. Retrieval is performed with all-MiniLM-L6-v2 sentence embedding model [Reimers and Gurevych, 2019].

We utilize the validation sets from Shaham et al. [2022] for QASPER and QUALITY, and use the prompts from Shaham et al. [2023]. For COQA, MUTUAL, and DIPLOMAT, we employ their validation splits and adapt the QUALITY prompt. Regardless of the dataset, we provide identical input prompts to both SLM and LLM to ensure consistent input processing costs. The output length is fixed in multi-choice datasets like QUALITY, and the brevity of responses in other datasets allows us to assume uniform output processing costs. We use greedy decoding (temperature 0) and draw a single sample for both the SLM and LLM. For self-verification, we use temperature=0.7 and draw 8 samples.

## C   Expanding AutoMix to Three-Models

A crucial use-case of AutoMix is to route between multiple models of varying scales in terms of cost and performance. In previous sections, we considered a two-model scenario, where SLM was 2 orders of magnitude smaller than the LLM. In this section, we evaluate how well AutoMix performs if we consider a third medium-sized language model (MLM). Specifically, we aim to address several questions: 1.) Does AutoMix perform better than baselines in the n-model scenario, 2.) Can AutoMix automatically learn to skip models if they are non-performant 3.) How beneficial is POMDP router in AutoMix, 4.) What are the failure cases of AutoMix in the 3-model scenario?

To answer these questions, we employ LLAMA2-13B/MISTRAL-7B/GPT-3.5 as the SLMs, LLAMA2-70B as the MLM, and GPT-4 as the LLM. We consider the following baselines:

- **FrugalGPT**: FrugalGPT proposed using a cascade style routing for n models, where the query is first sent to the SLM, and if the confidence is below a threshold, it is sent to the MLM, and so on. The thresholds are tuned on the validation set, so as to maximize the performance across different cost regions. Note that FrugalGPT trains $N - 1$ separate verifiers for each pair in a sequence of models. Similar to previous sections, we consider the cost of verifier as 0.

- **Union AutoMix**: Union AutoMix is a simple baseline, where we select between the two-model variants $AutoMix_{SLM-MLM}$, $AutoMix_{SLM-LLM}$ and $AutoMix_{MLM-LLM}$, depending on the cost requirements specified by the end-user. For instance, if the desired average cost is less than that of the MLM, $AutoMix_{SLM-MLM}$ may be employed, whereas $AutoMix_{MLM-LLM}$ or $AutoMix_{SLM-LLM}$ is utilized for cost regions exceeding that of the MLM. Precisely, for each cost region in validation set, the best-performing variant (highest $\Delta_{IBC}$) is chosen: $max_{\forall \text{cost region}}(Perf_{AutoMix_{SLM-MLM}}, Perf_{AutoMix_{MLM-LLM}}, Perf_{AutoMix_{SLM-LLM}})$. Note, the formula easily generalizes to larger $N$ by considering all possible pairs.

- **Chained AutoMix**: Chained AutoMix is a baseline where we chain the two-model variants $AutoMix_{SLM-MLM}$ and $AutoMix_{MLM-LLM}$. The query first goes to the SLM, and an $AutoMix_{SLM-MLM}$ decides between reporting the SLM answer or routing to the MLM. In the latter case, a second $AutoMix_{MLM-LLM}$ repeats the procedure using the MLM and LLM models.

## C.1 Extending IBC to N=3 models

In order to extend IBC metric to $> 2$ models, it is important to consider the cost region being considered. For instance, for $N = 3$ consider two separate cases: 1) when the SLM-MLM-LLM curve is convex, and 2) when the curve is concave. In the convex case, choosing between the MLM and SLM in low-cost regions is advantageous, while it is beneficial to choose between the MLM and LLM in high-cost regions. Accordingly, the suitable (more competitive) IBC curve is selected for evaluation accordingly. However, in the second case, when the IBC curves are concave, it would be more favorable to choose between the SLM and LLM and completely ignore the MLM, as in terms of incremental performance per cost, it consistently presents a disadvantage. Thus, the $\text{IBC}_{SLM-LLM}$ is chosen for evaluation throughout. Note that for general $N$ models, for each cost region, $N_2^C$ combinations need to be considered.

With the baselines, and evaluation metrics well defined, we next aim to answer each of the four answers mentioned above:

## C.2 Results of Automix with Three Models

### C.2.1 Does `AutoMix` outperforms baseline methods?

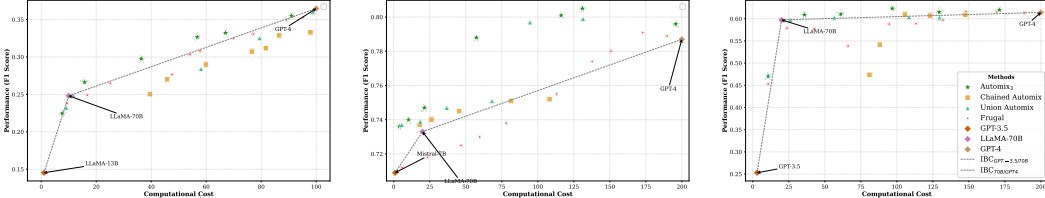

Figure 8: **AutoMix** outperforms baselines in $N = 3$-model scenario. Representative examples shows QASPER, DIPLOMAT, COQA datasets, where `AutoMix` consistently outperforms the baselines: `FrugalGPT`, Union `AutoMix`, Chained `AutoMix` across all cost regions considered.

Figure 8 presents results on three representative cases all from different datasets and SLMs. Across, all datasets, `AutoMix` consistently outperforms the baselines, `FrugalGPT`, Union `AutoMix`, and Chained `AutoMix`, across all cost regions considered. Since `FrugalGPT` and Chained `AutoMix` are cascade style routing, they are unable to directly route from SLM to LLM, and always first have to invoke the MLM, thus demonstrating poor performance. While Union `AutoMix` is able to alleviate this issue, it still needs to always invoke the MLM model and its costly Verifier to make a decision, while `AutoMix` is able to use information available from SLM verifier, to decide whether to route to MLM or LLM. Even in cases when routing to MLM takes place, POMDP in `AutoMix` leverages information from both SLM and MLM verifiers by updating its belief, providing it with a more nuanced decision-making capability.

### C.2.2 Can `AutoMix` skip distractor models?

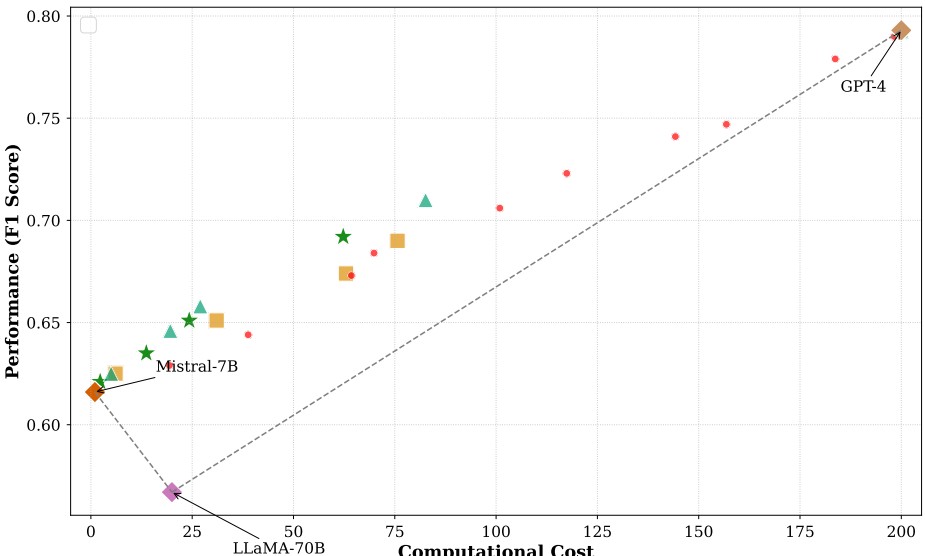

Figure 9: **AutoMix** automatically skips distractor MLM models that underperforms the SLM. However, `FrugalGPT` and Chained `AutoMix` struggle to discern this, resulting in poor performance across the board. Results are on MUTUAL dataset with MISTRAL-7B as SLM.

As a robustness, we now consider a scenario, where MLM underperforms the SLM. Such a scenario is common in practice since models come from different families, trained on different kinds of data, and it is possible that one model is better than the other despite the scale. Figure 9 considers two/three scenarios, where MLM underperforms the SLM. Note that in such cases, `FrugalGPT` performs significantly worse since it always routes to the MLM for higher cost regions, thus wasting resources. Thus `FrugalGPT` requires an extra consideration on validation set, to weed out such distractor models. However, `AutoMix` is able to learn the irrelevance of MLM, and learns to route directly from SLM to LLM, as needed. The performance of `AutoMix` is similar to Union `AutoMix` in such cases since the MLM is never used.

### C.2.3 Should Low Performing MLMs be ignored?

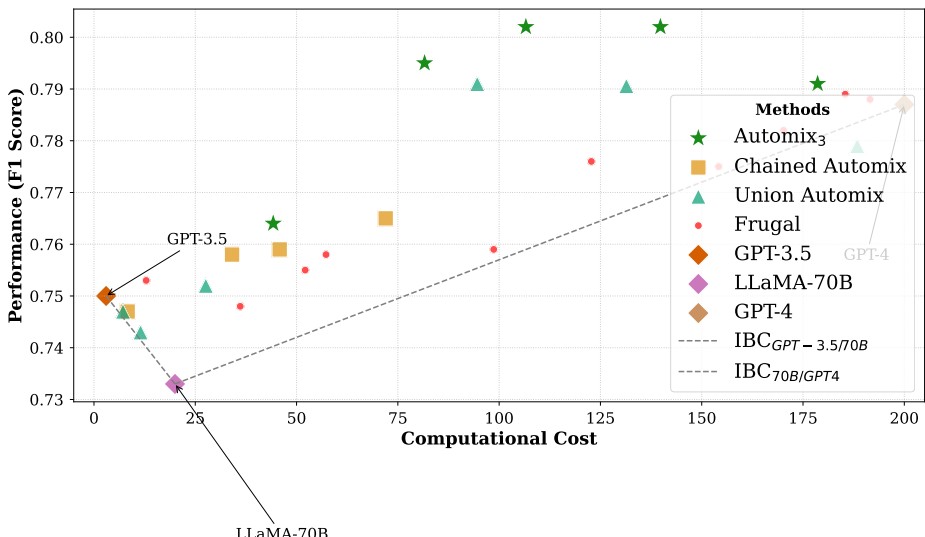

Figure 10: `AutoMix` can leverage even worse performing MLM by identifying questions, where MLM is better than SLM and making use of MLM veirififcation probabilities. All other baselines, are unable to leverage this information, and perform worse than `AutoMix`

While in the previous section, we established that low-performing MLMs should be ignored, this may not always be the case. This is because there may be certain types of questions where MLM is indeed better than SLM, and if, through self-verification probabilities, it is possible to exploit such information, it can further improve the cost-quality of `AutoMix`. Figure 10 demonstrates one such case. Specifically, `AutoMix` identify that low verification probabilities on SLM implies that MLM is likely more better than SLM. Thus, it routes to MLM models appropriately. Further, the combined verification information from SLM and MLM, allows `AutoMix` to make a more nuanced decision, and thus outperforms all other baselines considered. So much so, `AutoMix` is able to identify patterns that result in statistically overperforming even the LLM even at roughly half the cost. The experiment, along with the previous ones, demonstrates the efficacy and usefulness of the POMDP router in `AutoMix`, as it is automatically able to handle a wide variety of cases and consistently provides the best cost-quality tradeoff.

### C.2.4 Does `AutoMix` always provide significant gains?

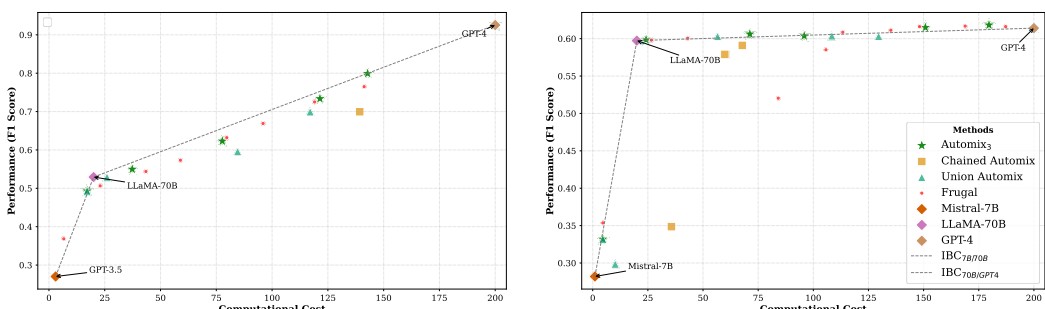

Figure 11: `AutoMix` perform similarly to baselines without any statistically significant gains in the two representative examples, owing to relatively poor self-verification.

In previous subsections, we considered cases, where `AutoMix` was significantly better than `FrugalGPT` and often the other variants of `AutoMix`. However, in this section, we consider a couple of cases, where `AutoMix` do not provide significant improvements. Figure 11 represents the two cases. Specifically, we note that in both cases, `AutoMix` is similar to `FrugalGPT`. Further, the advantage over IBC lines is negligible or even negative. We qualitatively analyze and find that the self-verification by MLM in such scenarios is not particularly useful, and thus, POMDP is not able to provide significant gains. We believe future work should address such challenges by considering contextual information or other forms of verification to improve the performance of our methods in such cases.

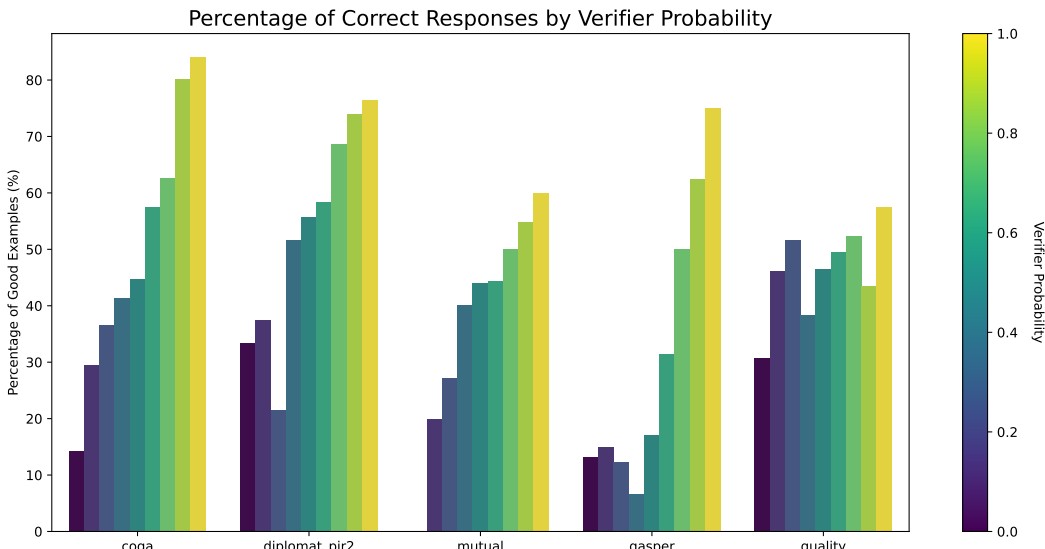

Figure 12: **Verifier Probability and Correctness:** Percentage of correct responses across distinct verifier probability bins for LLAMA2-13B as SLM. Each bin represents a range of verifier probabilities and the corresponding accuracy of the responses within that probability range across various datasets. Notably, for all datasets, excluding QUALITY and QASPER, a higher verification score generally corresponds to a larger proportion of correct examples, indicating that the verifier is, to an extent, capable of discerning the reliability of responses generated by itself. We use a router behaving like a meta-verifier to get around these noisy predictions.

Figure 13, 14, 15, 16, 17 shows results for $N = 3$ scenario for `AutoMix` along with baselines.

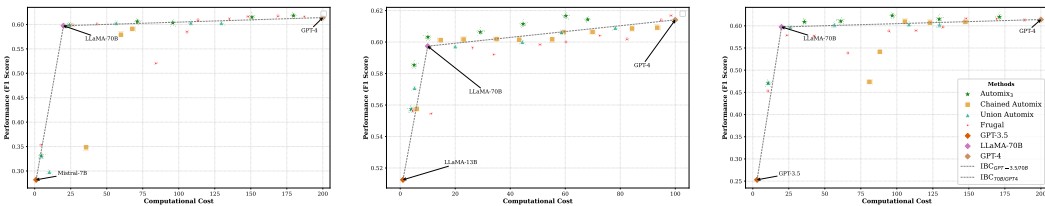

Figure 13: **AutoMix** in $N = 3$-model scenario. Dataset: COQA.

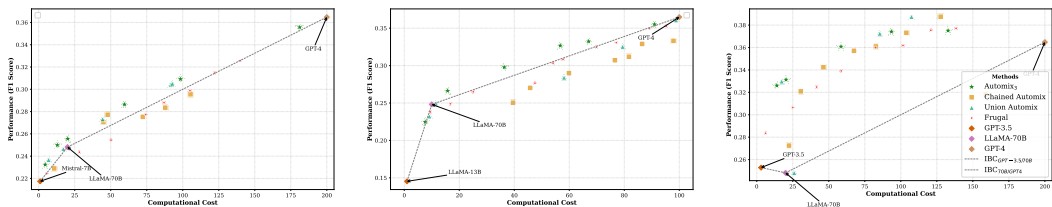

Figure 14: **AutoMix** in $N = 3$-model scenario. Dataset: QASPER.

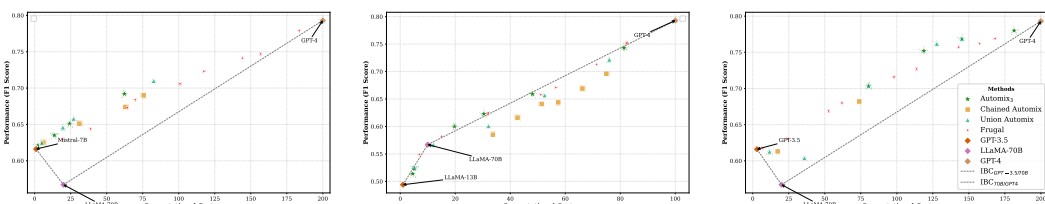

Figure 15: **AutoMix** in $N = 3$-model scenario. Dataset: QUALITY.

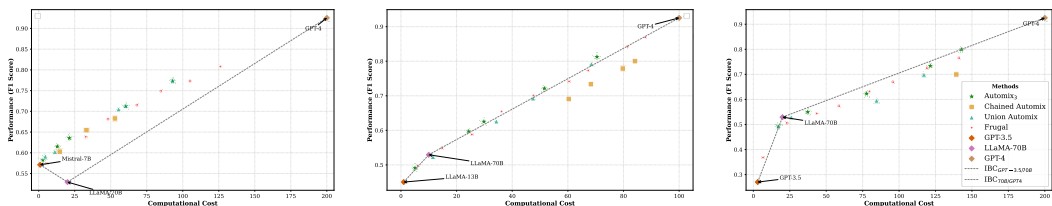

Figure 16: **AutoMix** in $N = 3$-model scenario. Dataset: MUTUAL.

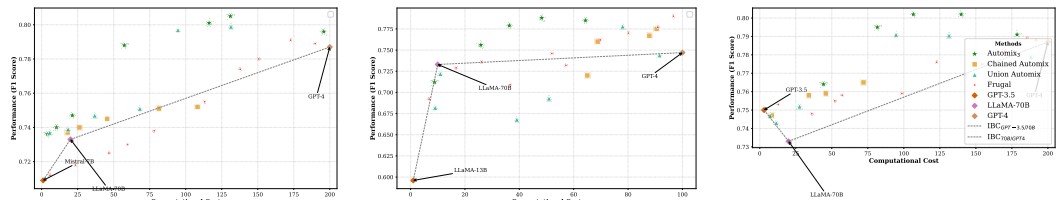

Figure 17: **AutoMix** in $N = 3$-model scenario. Dataset: DIPLOMAT.

# D  Additional Results

## D.1  Main Results

Figure 19, 18, 4 shows the results of AutoMix for 3 different choices of SLM. Notice, for 13/15 configurations, AutoMix-POMDP is clearly better than all other baselines throughout different operating costs. Even in the remaining 2 cases, AutoMix is competitive and almost equivalent to the baselines. We notice similar trends from Section 5.3 for LLAMA2-13B and GPT-3.5 as well.

## D.2  **AutoMix** is effective across different Cost-Ratios

In this section, we furhter compare how, AutoMix compare to baselines for different cost-ratios. Figure 20 demonstrates that at the API based cost ratio, AutoMix outperform all baselines. Even at a 20:1 ratio (an unlikely, pessimistic scenario for AutoMix), AutoMix outperforms baselines, albeit with smaller margins. Conversely, at a 2000:1 ratio, AutoMix shows even higher improvements. The result demonstrate the robustness of AutoMix to ever-changing costs of models and APIs.

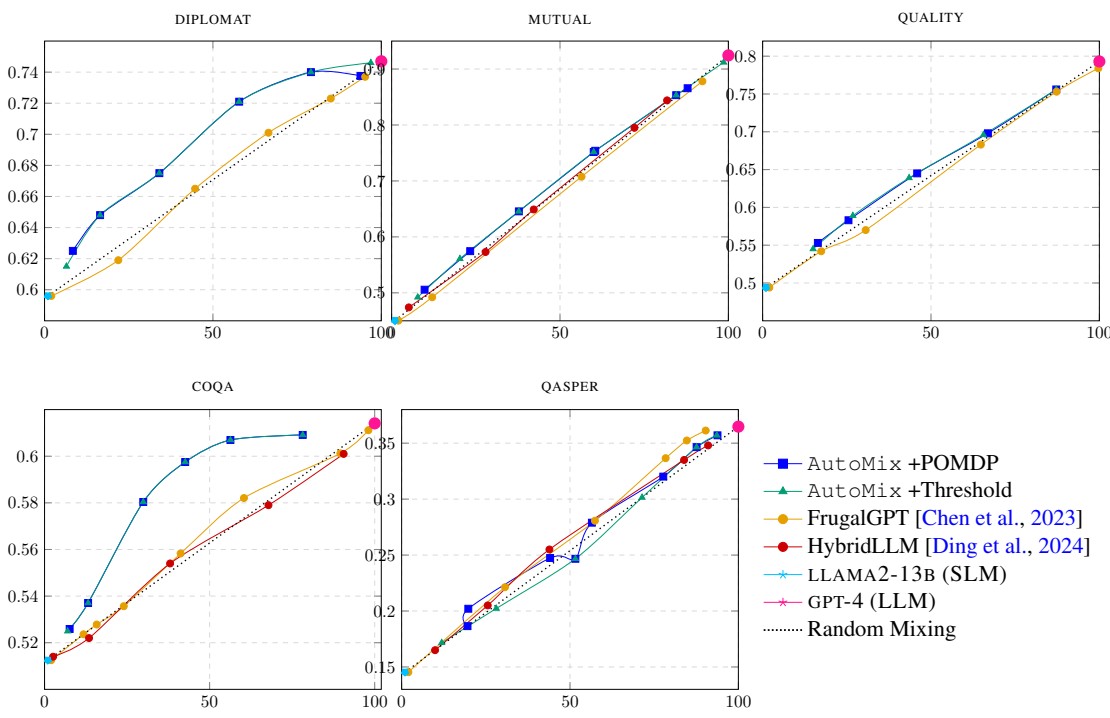

Figure 18: **Main Results:** performance **(y-axis)** vs. cost **(x-axis)** for different methods on the small and large LLAMA2-13/GPT-4. POMDP based meta-verifier is consistenly above the linear interpolation of SLM-LLM, signifying a higher incremental benefit per unit cost (IBC).

### D.3 Out-of-Domain Generalization of `AutoMix`

The POMDP router introduced in `AutoMix` does not assume any specific task or dataset characteristics to work, as it relies only on self-verification confidence as input. Thus, our POMDP router is generalizable to various datasets and tasks.To further demonstrate this, we evaluate out-of-domain generalization by training on one dataset and evaluating on others. Specifically, we train the router on one of the five datasets and evaluate it on the other four. We repeated the experiment for all five datasets and three SLMs. The results in different settings, as shown in the Table 5, indicate that our POMDP router consistently outperforms both FrugalGPT and HybridLLM. The design of the POMDP router, along with these results, highlights the strong generalizability of our proposed method.

|  | Mistral-7b | LLama-13b | GPT-3.5 |
|---|---|---|---|
| Automix | **28.3** | **31.5** | **70.9** |
| Frugal | 12.5 | 0.0 | 14.3 |
| Hybrid | 2.4 | -2.8 | 7.6 |

Table 5: Out-of-domain generalization across various SLMs for different methods. Scores are averaged across five datasets. `AutoMix` significantly outperforms other methods.

## E   Few-Shot Prompts

### E.1   Verifier Prompts

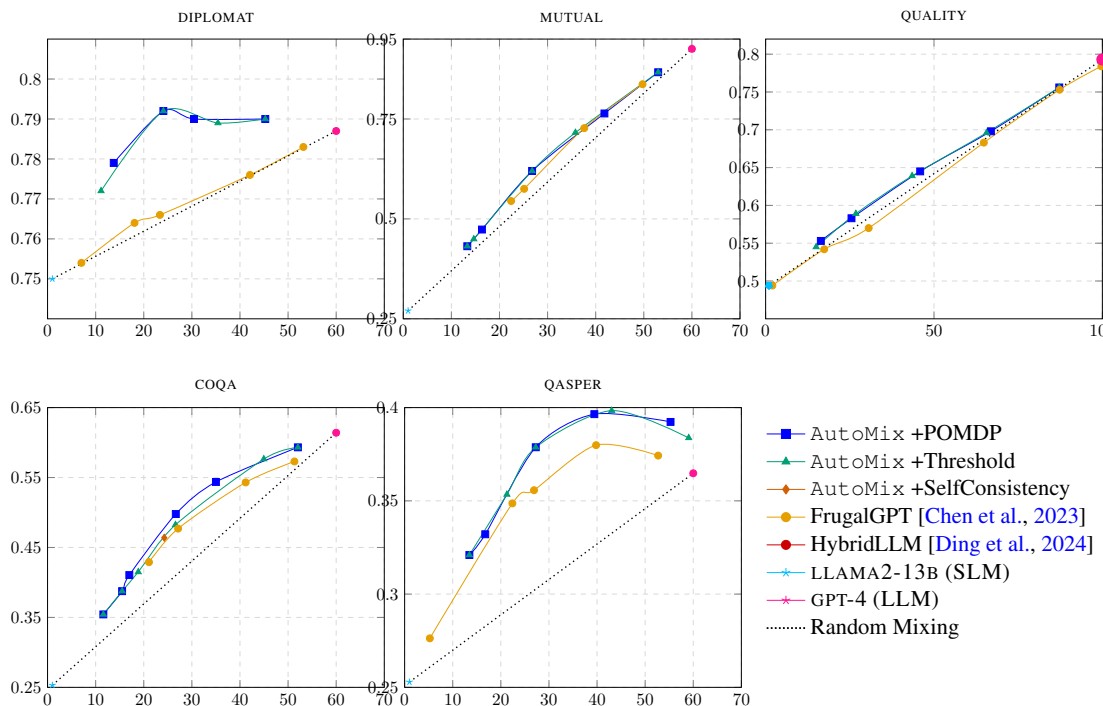

Figure 19: **Main Results:** performance **(y-axis)** vs. cost **(x-axis)** for different methods on the small and large GPT-3.5/GPT-4. POMDP based router is consistently above the linear interpolation of SLM-LLM, signifying a higher incremental benefit per unit cost (IBC). `AutoMix` is the best performing method consistently across all datasets, often by large margins.

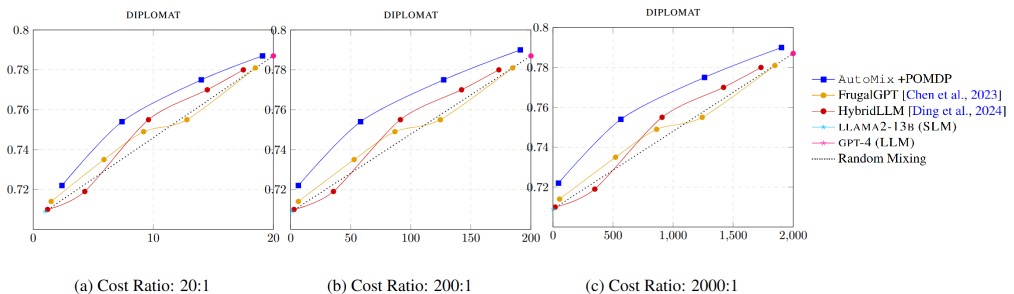

(a) Cost Ratio: 20:1    (b) Cost Ratio: 200:1    (c) Cost Ratio: 2000:1

Figure 20: We compare `AutoMix` to baselines across various LLM to SLM cost ratios. For all considered cost-ratios, `AutoMix` outperform baselines with varying degree of margins.

```
Story:
{relevant parts of the story}

{instruction}

Question: {question}

Answer:
```

Listing 2: **Task Prompt.** We experiment with long-context reasoning tasks, which require answering questions from stories, legal contracts, research papers, and novels.

```
Context: {context}

Question: {question}

AI Generated Answer: {generated_answer}

Instruction: Your task is to evaluate if the AI Generated Answer is
↪  correct, based on the provided context and question. Provide the
↪  judgement and reasoning for each case. Choose between Correct or
↪  Incorrect.

Evaluation:"'
```

Listing 3: **Verification Prompt.** The verification process is framed as a natural language entailment task, where the model determines the validity of the model-generated answer with respect to the context and question.

```
Context: The manuscript, discovered in 1980 in a dusty attic, turned out
↪  to be a lost work of Shakespeare.\n
Question: Whose lost work was discovered in a dusty attic in 1980?\n
AI Generated Answer: Shakespeare\n
Instruction: Your task is to evaluate if the AI Generated Answer is
↪  correct, based on the provided context and question. Provide the
↪  judgement and reasoning for each case. Choose between Correct or
↪  Incorrect.\n
Evaluation: The context specifically mentions that a lost work of
↪  Shakespeare was discovered in 1980 in a dusty attic.

Verification Decision: The AI generated answer is Correct.

---

Context: The celestial event, known as the Pink Moon, is unique to the
↪  month of April and has cultural significance in many indigenous
↪  tribes.\n
Question: In which month does the celestial event, the Pink Moon,
↪  occur?\n
AI Generated Answer: July\n
Instruction: Your task is to evaluate if the AI Generated Answer is
↪  correct, based on the provided context and question. Provide the
↪  judgement and reasoning for each case. Choose between Correct or
↪  Incorrect.\n
Evaluation: The context clearly states that the Pink Moon is unique to
↪  the month of April.

Verification Decision: The AI generated answer is Incorrect.

---

{truncated examples}

Context: {context}\n
Question: {question}\n
AI Generated Answer: {generated_answer}

Instruction: Your task is to evaluate if the AI Generated Answer is
↪  correct, based on the provided context and question. Provide the
↪  judgement and reasoning for each case. Choose between Correct or
↪  Incorrect.

Evaluation:
```

Listing 4: **Few-Shot Verifier Prompts:** 3-shot verifier prompt for evaluating the correctness of SLM's answer. The same prompt is used for all datasets.

