# OpenReview forum: "AutoMix: Automatically Mixing Language Models"
_NeurIPS.cc/2024/Conference — NeurIPS 2024 poster_

### Official Review · Reviewer_eJtm · 2024-06-17

**Soundness:** 3
**Presentation:** 3
**Contribution:** 3
**Rating:** 6
**Confidence:** 4

**Summary:**

This paper investigates how to automatically and adaptively select the LLM with a smaller size but keep good performance so that it offers the potential to achieve a better cost-effectiveness trade-off. This work uses a smaller model to predict first and then uses the small model to do self-verification. If more computation is required, a POMDP-based router can effectively select an appropriately sized model. The proposed approach can reduce around 50% cost as reported in the paper.

**Strengths:**

1) Adaptively selecting model sizes is smart and cool. Using a smaller model first makes sense because it can reduce the overhead when the task is difficult.
2) The paper is very well-written and easy to understand. The Verifier Qualitative Analysis in the appendix is also helpful.

**Weaknesses:**

1) It is good that this work provides a detailed description of their cost modeling. However, it seems that there are many assumptions in it. How can authors justify that this is not biased?
2) More importantly, this work requires more discussion about the real-world throughput  (response latency). Although the computation cost is reduced, the user experience would become worse if the latency is significant larger.

**Questions:**

See Weakness.

---

> ### Author Rebuttal · Authors · 2024-08-07
>
> Thank you for taking the time to review the paper and provide feedback! We believe all your questions can be addressed within this discussion period, but we would love to provide further clarification if needed.
>
> ---
>
> ### It is good that this work provides a detailed description of their cost modeling. However, it seems that there are many assumptions in it. How can authors justify that this is not biased?
>
> **Our cost modeling directly reflects the real-world pricing of APIs, as detailed in Appendix D (Lines 649-657).** We did not introduce any subjective assumptions in the cost ratios. Furthermore, to handle evolving cost dynamics of LLM API providers, we conduct analysis with varying cost ratios in Section 5.4 (Figure 5, Right), confirming that AutoMix offers significant improvements even when LLM costs are reduced by a factor of 2-4. Our method is robust across a wide range of cost ratios of SLMs and LLMs. This robustness indicates that our results are not biased by the chosen cost model.
>
>
> ---
>
> ### More importantly, this work requires more discussion about the real-world throughput (response latency). Although the computation cost is reduced, the user experience would become worse if the latency is significantly larger.
>
> We thank the reviewer for bringing up this point. As we note in Section 5.4, AutoMix is agnostic to the specific nature of the cost metric. Latency can be modeled similarly to how the cost is modeled, with the larger model incurring higher latency. As noted in our limitations, “AutoMix can automatically handle latency as another form of cost, but it might lead to less significant improvements” owing to lower latency differences between SLMs and LLMs. However, the exact latencies vary by different providers. As a real-world example, an ~8B model, through API providers such as Groq, provides generation speeds up to 1000 tokens/s, compared to roughly 30 tokens/s for GPT-4. In such scenarios, the latency of routing to LLMs would be far higher than a fixed number of more calls to SLMs for reaching a given performance. Therefore, AutoMix can improve latency for a given performance, especially by reducing the higher latency of calls to LLMs.

---

> > ### Comment · Reviewer_eJtm · 2024-08-11
> >
> > Thanks for the reply. I will keep my score as it has been relatively positive.

---

> > > ### Author Response · Authors · 2024-08-11
> > >
> > > Thank you for acknowledging our response. We look forward to address any additional concerns you may have during the remainder of the discussion period!

---

### Official Review · Reviewer_RAHf · 2024-07-01

**Soundness:** 2
**Presentation:** 3
**Contribution:** 2
**Rating:** 5
**Confidence:** 4

**Summary:**

This paper presents AutoMix, a method that routes query to language models (LMs) of various size and capabilities to optimize the performance within a cost budget. AutoMix has two key technical features, a few-shot self-verification mechanism which estimates the reliability of its own outputs, and a POMDP-based router that can select a model to optimize the reward. Experiments across five LMs and five datasets show that AutoMix outperform two baselines (i.e., FrugalGPT, HybridLLM).

**Strengths:**

1. Formulating the query routing optimization problem as a Partially Observable Markov Decision Process (POMDP) is interesting. Treating performance of different models as $S$ is quite smart since the most challenging part of the optimization problem is we want to make as less computation as possible while we cannot observe whether an LM's output is correct or not since the optimization happens in the testing stage. The property of the optimization problem matches very well with POMDP.
2. The experimental results show that AutoMix outperforms FrugalGPT and HybridLLM which are two recent methods in this setup.

**Weaknesses:**

1. The experimental setup is not very rigorous. The major metric in this paper is $\Delta_{IBC}$ which will be influenced by how you set the cost of different LMs in your experiment. Although the arguments provided in Line 216-219 is legitimate, setting the cost as 1, 60, 100, 200 respectively feels like setting "hyperparameters" to me. Given the results on different datasets and models in Table 1 are not consistent, I don't have confident with the conclusion from the current results. For example, will the trend change if I set the cost as 10, 60, 100, 200?
2. While I praise the integration of POMDP as interesting and well-motived, I am not sure whether such integration adds much to the performance given the main results in Figure 4. It seems like "AutoMix+POMDP" and "AutoMix+Threshold" achieve similar results while the latter excels in simplicity. I am also wondering how much adding POMDP will increase the overhead since the system needs to run the AdaOps POMDP solver.
3. The explanation of "domain-specific training" could be clearer. Since this phrase in LM literature usually refers to post-train LM on domain-specific data, it's necessary to clearly define what it means in your method. Although I believe I understand the training of router in your method, I have a clarification question: Does computing `P(o|s)` require running the inference on all the samples in the dataset and also running the self-verification step? If this is the case, should this part of the cost be considered?

**Update:** Some of these points are addressed by the author response, so I updated the score from 4 (Borderline Reject) to 5 (Borderline Accept).

**Questions:**

See my comments in "Weaknesses".

**Limitations:**

The paper has a "Limitations" section.

---

> ### Author Rebuttal · Authors · 2024-08-07
>
> Thank you for taking the time to review the paper and provide feedback! We believe all your questions can be addressed within this discussion period, but we would love to provide further clarification if needed.
>
> ---
>
> ### Although the arguments provided in Line 216-219 is legitimate, setting the cost as 1, 60, 100, 200 respectively feels like setting "hyperparameters" to me.
>
>
> We would like to address the concerns regarding cost values (ratios) below:
> 1. *Cost values as hyperparameters:* We respectfully disagree with the reviewer that cost is a hyperparameter for our approach. Cost values signify the parameters of the problem. Moreover, these parameter values are not in any way optimized by our system but motivated by the cost of apis of popular and SOTA language models. Nevertheless, since these values could also change in future, we measure robustness and sensitivity of our approach if we vary cost ratio between different SLMs and LLMs in Section 5.4 (as described below).  To reiterate, we use conservative estimates to the pricing: 30 USD/M tokens for GPT-4, 0.1/M for Mistral7B, 0.225/M for Llama-13b, and 0.5/M for GPT3.5, giving cost-ratios of roughly 200, 100 and 60
> 2. *Robustness of AutoMix to different cost scenarios:* To understand the effect of different cost values, in section 5.4, we further evaluated the performance of Automix with different cost values. As we describe in the paper, a huge cost ratio between LLM and SLM would be favorable to Automix, as relative cost of self-verification decreases, while lower cost would be detrimental. Nonetheless as stated in lines (267-269), “The results suggest that even for a ten times lower cost ratio, AutoMix begins to deliver good gains for many datasets” and outperforms all baselines at even half the cost ratio. Further, incase of lower SLM costs (a practical scenario as described in lines 221-223), Automix performs even better than the scores reported in main results. To further, supplement the analysis, we also include cost-performance curves for different cost-ratios in Figure 2 of general response.
> 3. Thirdly, these costs values and methods should be seen beyond the pricing of LLMs. These could signify latency or energy costs of models.
>
> We believe beyond just the exact cost values, Automix provides a novel method to balance cost performance trade off which works across a wide variety of cost ranges and is robust across a very wide range of cost ratios. In practical scenarios, it performs significantly better than other baselines.
>
>
>
>
>
>
> ---
>
> ### While I praise the integration of POMDP as interesting and well-motivated, I am not sure whether such integration adds much to the performance given the main results in Figure 4.
>
> POMDP is a general setup, essential for generalizing to scenarios with more than two models, and sets a very robust baseline for future endeavors consisting of more complicated routing strategies. Further, as we discuss in Section 5.2 simpler methods perform much poorly than a POMDP model for settings with more than two models, as demonstrated in Figure 7. In addition to Figure 7, we include more such cases in Figure 1, general response, demonstrating the importance of POMDP.
>
> Finally, we note that implementing the POMDP model is straightforward and fast, requiring only a few lines of code (~5) for the end-user to incorporate our library. Additionally, as mentioned in line 285, it takes less than 1 ms to run the POMDP model for an input query. We will include this discussion in the revised paper to better justify our use of POMDPs.
>
>
> ---
>
> ### The explanation of "domain-specific training" could be clearer. Since this phrase in LM literature usually refers to post-train LM on domain-specific data, it's necessary to clearly define what it means in your method.
>
> “Domain-Specific training” refers to the usual meaning in LM literature of “post-training LM on domain-specific data” We note, that unlike FrugalGPT and HybridLLM, Automix does not require domain-specific training of **verifier**
>
> ---
>
> ### Although I believe I understand the training of router in your method, I have a clarification question: Does computing P(o|s) require running the inference on all the samples in the dataset and also running the self-verification step? If this is the case, should this part of the cost be considered?
>
> Yes, computing P(o|s) requires running inference on all dataset samples and the self-verification step. However, note that this **cost is incurred only during training and not evaluation**. Further, even the **training cost is similar to baselines** since both FrugalGPT and HybridLLM require running the inference on all samples, and additionally requires training verifiers. While Automix requires running self-verification, verification is run only on smaller models, which are usually significantly cost-efficient than LLMs (eg, 60-200 times cheaper), thus adding negligible training costs. We will clarify this clearly in the revised version.

---

> > ### Comment · Reviewer_RAHf · 2024-08-10
> >
> > Thanks for the very detailed response!
> >
> > > We believe beyond just the exact cost values, Automix provides a novel method to balance cost performance trade off which works across a wide variety of cost ranges and is robust across a very wide range of cost ratios.
> >
> > It's worth highlighting this in the paper as the first-time reader may fail to realize this when looking at the main results.
> >
> > > Finally, we note that implementing the POMDP model is straightforward and fast, requiring only a few lines of code (~5) for the end-user to incorporate our library. Additionally, as mentioned in line 285, it takes less than 1 ms to run the POMDP model for an input query.
> >
> > This removes my doubt.
> >
> > > However, note that this cost is incurred only during training and not evaluation. Further, even the training cost is similar to baselines since both FrugalGPT and HybridLLM require running the inference on all samples.
> >
> > Please include this in the paper.
> >
> > Given major points are addressed, I updated my score to indicate a weak support of the acceptance.

---

> > > ### Author Response · Authors · 2024-08-11
> > >
> > > Thank you for considering our response and raising our score! We are glad to know that all major points have been addressed. We will add the discussed clarifications in the revised paper.
> > >
> > > We are happy to address any further questions you may have!

---

### Official Review · Reviewer_NEfd · 2024-07-12

**Soundness:** 3
**Presentation:** 3
**Contribution:** 3
**Rating:** 6
**Confidence:** 3

**Summary:**

This work introduces a novel solution called AutoMix to achieve an optimal balance between performance and cost when using various scales of large language models (LLMs). The paper presents two variants of AutoMix: AutoMix-T, which employs a thresholding strategy, and AutoMix-P, which uses a POMDP-based strategy. Additionally, a new metric, IBC, is proposed to measure cost-performance efficiency. Experimental results on five datasets and a two-model setting with three LLMs demonstrate the superiority of AutoMix compared to other baselines.

**Strengths:**

1. This work addresses an intriguing problem: how to leverage small language models as assistants to reduce costs while maintaining comparable performance when using large language models is expensive.
2. The proposed solution is technically sound and well-described.
3. The empirical analysis demonstrates that the proposed method outperforms the baselines.

**Weaknesses:**

1. There is a lack of analysis regarding absolute performance decay. In most cases, performance is more critical than cost. The authors should show the impact on absolute performance when applying AutoMix.
2. The POMDP-based router appears to introduce additional learning costs; however, this aspect is not analyzed.
3. As noted by the authors, the generalizability of AutoMix is not well demonstrated, as the experiments cover only limited settings.

**Questions:**

1. Generally, the performance of small language models (SLMs) is inferior to that of large language models (LLMs). Adopting the outputs of SLMs as the final response may result in considerable performance decay. Could you provide a quantified analysis of this impact?

2. Could you provide a detailed analysis of how the POMDP-based router influences the decision of when to call LLMs? Clearly, the POMDP-based method cannot provide perfect predictions.

**Limitations:**

No.

---

> ### Author Rebuttal · Authors · 2024-08-07
>
> Thank you for taking the time to review the paper and provide feedback! We believe all your questions can be addressed within this discussion period, but we would love to provide further clarification if needed.
>
> ---
>
> ### There is a lack of analysis regarding absolute performance decay.
>
> We would like to clarify that Figures 4, 13, and 14 exactly show how the absolute performance varies as the cost changes. Specifically, our task is a multi-variable optimization problem, where there is a trade-off between cost and performance. Thus, performance can be compared only at fixed costs. Across all dataset and model settings, the figures show that when evaluated at the same costs, our method has higher performance than any of the baselines. It should also be noted that while higher performance is desirable when possible, many applications aim to trade off due to cost/latency concerns. So, we believe optimizing for this trade-off is well motivated and an important concern from a practical perspective when deploying these models to many real-world applications.
>
> ---
>
>
> ### POMDP-based router appear to introduce additional learning costs; however, this aspect is not analyzed.
>
> The **POMDP-based router does not introduce additional learning costs**. The POMDP router is computationally efficient, requiring less than 1 second to train on a single CPU thread. This is substantially less than the training time required by routers in FrugalGPT and Hybrid LLM, which necessitate GPU resources and longer training durations. We will clarify this point in the revised paper.
>
> ---
>
> ### As noted by the authors, the generalizability of AutoMix is not well demonstrated, as the experiments cover only limited settings.
>
> We would like to clarify our limitations here. While our experiments were conducted on context-grounded reasoning datasets, they covered a wide range of settings (15 in total: 3 models x 5 datasets), including different domains, objectives, answer types, and SLM, LLM performances, and their performance gaps. AutoMix consistently outperformed all baselines across these varied settings, demonstrating its broad applicability. Furthermore, our POMDP router assumes access to only self-verification probabilities and is thus generalizable to any setting or domain of task.
>
>
> ---
>
> ### Generally, the performance of small language models (SLMs) is inferior to that of large language models (LLMs). Adopting the outputs of SLMs as the final response may result in considerable performance decay. Could you provide a quantified analysis of this impact?
>
> We acknowledge that owing to scaling laws in LLMs, SLMs have inferior performance compared to LLMs but at the same time, the performance gains from LLMs comes at additional cost/latency. Using LLMs for every problem could be an overkill for easy problems(where SLM could solve) and problems which are very hard (which even LLMs can’t solve). Our approach aims to exploit these gaps by using a smart self-verification and POMDP routing where one can achieve considerably higher performance for a given fixed cost. We believe this is well motivated from a practical standpoint of deploying language models in the real world where billions of queries are passed to language models and one also needs to optimize for cost in addition to performance. **Our cost-performance trade off clearly demonstrates this in Figure 4, 13 and 15.**
>
>
> ---
>
> ### Could you provide a detailed analysis of how the POMDP-based router influences the decision of when to call LLMs? Clearly, the POMDP-based method cannot provide perfect predictions.
>
> We understand that any ML mechanism like POMDP cannot provide totally perfect predictions. However, it is crucial to note that the POMDP router is able to take advantage of self-verification probabilities to avoid routing to LLMs in primarily two cases: a) where the query is easy such that both SLM and LLM would give a correct answer, and b) the hard queries where both SLM and LLM would be wrong. This could be much more difficult to determine than a simple thresholding baseline. For instance, in the Qasper dataset with Mistral-7B as SLM, POMDP understands that lower confidence values correspond to such cases, and while other methods would have routed to LLM, POMDP returns the SLM answer, saving cost. Further, in a >2 model setup, it can make more nuanced decisions, for instance using combined information from SLM and MLM verifier, as demonstrated in Figure 6 in the main paper, and Figure 1 in general response.

---

> > ### Author Response · Authors · 2024-08-12
> >
> > Thank you again for your valuable review. If you have any further questions or concerns, please let us know so we can address them before the end of the discussion phase. If you feel that our responses have addressed your original concerns, please consider updating your evaluation. We appreciate your time and effort in this process!

---

> > > ### Comment · Reviewer_NEfd · 2024-08-13
> > >
> > > Thanks for the detailed response. It addressed my main concerns, and I raised my score.

---

### Official Review · Reviewer_E2SA · 2024-07-13

**Soundness:** 3
**Presentation:** 3
**Contribution:** 3
**Rating:** 6
**Confidence:** 4

**Summary:**

AutoMix is an approach designed to optimize the performance and computational cost of LLMs by selecting the appropriate model based on the difficulty of the task. This is achieved through a few-shot self-verification mechanism and a Partially Observable Markov Decision Process (POMDP) based router. The few-shot self-verification estimates the correctness of outputs from a smaller LM, and the POMDP router decides whether to route the query to a larger LM. Experiments demonstrate that AutoMix can reduce computational cost by over 50% while maintaining performance across various datasets and models.

**Strengths:**

- The author introduce a novel combination of few-shot self-verification and POMDP-based routing to optimize the use of large language models. It is very thoughtful to make SLM itself both the answerer and the judge, and to consider it as the first candidate when selecting a suitable model. And the author successfully uses POMDP to meet the need of selecting a suitable model.
- They provide a reasonable metric for cost-efficiency analysis. When the cost grows or when the performance decreases, the results of the metric will become small.
- The methodology is sound, and the results demonstrate the effectiveness of AutoMix in reducing computational costs by over 50% while maintaining same performance. The experiments are robust, and the comparisons with several baselines prove the advantages of AutoMix.

**Weaknesses:**

- While the paper demonstrates the effectiveness of AutoMix across various datasets, it primarily focuses on dialogue and context-grounded reasoning tasks. It is not clear how well the approach would generalize to other types of tasks, such as factual question answering or commonsense reasoning.
- The paper mentions that the POMDP can learn from as few as 50 examples, but it does not elaborate on the conditions or types of data required for effective training.
- Although the paper compares AutoMix with strong baselines like FrugalGPT and HybridLLM, the comparisons could be more detailed in terms of why AutoMix outperforms these baselines in specific cases.

**Questions:**

I think it would be interesting to just randomly pick up a LLM for each question in dataset, and to see the total cost and the quality of results. Have anyone done this before? Because there might be a case where, even without any filtering strategy, just putting several models with different performance and different cost out there is itself effective. And I think may be it’s possible to derive it mathematically？

**Limitations:**

The author pointed out the limitations  in the discussion.

---

> ### Author Rebuttal · Authors · 2024-08-07
>
> Thank you for taking the time to review the paper and provide feedback! We believe all your questions can be addressed within this discussion period, but we would love to provide further clarification if needed.
>
> ---
>
> ### Q: I think it would be interesting to just randomly pick up a LLM for each question in dataset, and to see the total cost and the quality of results. Have anyone done this before?
>
> We thank the reviewer for raising the point. As noted in our paper (Lines 199-201), the dotted IBC line in Figure 4, “signifies the cost-performance curve that one would obtain if each data point was routed randomly between SLM and LLM.” We denote this baseline as “Random Mixing” in main results (Figure 4).
> All our comparisons are done with respect to this baseline, and we demonstrate that while FrugalGPT and HybridLLM often struggle to beat this simple baseline, Automix consistently outperforms across all datasets and models.
>
> ---
>
> ### The paper mentions that the POMDP can learn from as few as 50 examples, but it does not elaborate on the conditions or types of data required for effective training.
>
> **We make no assumptions on the condition or type of data required for learning POMDP.** Our method does not require specific conditions or types of data for POMDP training. We evaluated it across 5 datasets and 3 SLMs (totaling 15 settings), encompassing a wide variety of domains and task types (e.g., reasoning, next utterance prediction, QA), consistently demonstrating strong performance.
>
> ---
>
> ### The comparisons could be more detailed in terms of why AutoMix outperforms these baselines in specific cases.
>
> We note that our Self-Verification mechanism and POMDP router collectively contribute to the superior performance of AutoMix. For instance, in the Diplomat dataset with Mistral-7B as the SLM, our method achieved significantly higher self-verification accuracy compared to FrugalGPT and Hybrid LLM.
>
> Further, unlike previous methods, POMDP automatically identifies cases where the query is very complex such that both SLM and LLM would give an incorrect answer. In such cases, while the previous methods would still route to larger LLM, POMDP would report SLM’s answer, saving cost. Further, in more than >2 model setup, POMDP can make more nuanced decisions, for example by considering the combined information from both SLM and MLM verifier. We will make this discussion clearer in the revised paper.
>
> ---
>
> ### It is not clear how well the approach would generalize to other types of tasks
>
> 1. We thank the reviewer for bringing up the interesting point. We evaluate our method on a variety of models and datasets related to comprehension QA, and dialogue reasoning, which themselves offer a great diversity, such as answering questions from research papers (testing model's *factuality*), *reasoning* questions in multi-turn dialogue, next utterance prediction, conversational QA. The datasets are of varying difficulty, with LLM performance ranging from 35% to 95%, and different output formats: MCQs and open-ended generation. Despite the diversity, automix consistently outperforms baselines, demonstrating its generalizability. Further, our POMDP formulation is domain-agnostic and automatically extensible to other task types.
>
>
> 2. Moreover, as we have already note in our limitations: “AutoMix is designed with a dialogue-related or context-grounded reasoning setup in mind for effective self-verification” we note that the inability to do self-verification for reasoning tasks is a limitation of current LLMs [1, 2,3, 4] rather than our method. On the contrary, our work demonstrates how to successfully use any self-verification for context-grounded reasoning tasks using a suitable routing function. Therefore, we consider our contribution to context-grounded reasoning tasks to be a strength.
>
> 3. Further, we evaluate our method on an *additional common-sense reasoning dataset*: CICERO [5]. We find automix outperforms baselines by more than absolute 30% points. The results demonstrate potential superiority of Automix on datasets not evaluated in our paper. This additional result is provided in Table 2 of the general response.
>
> |          | Mistral-7b |
> |----------|------------|
> | Automix  | **66.4**   |
> | Frugal   | 32.1       |
> | Hybrid   | 19.7       |
>
>
> ---
>
>
>
>
> **References**
>
> Huang, J., Chen, X., Mishra, S., Zheng, H.S., Yu, A.W., Song, X., & Zhou, D. (2023). Large Language Models Cannot Self-Correct Reasoning Yet. ArXiv, abs/2310.01798.
>
> Tyen, G., Mansoor, H., Chen, P., Mak, T. and Cărbune, V., 2023. LLMs cannot find reasoning errors, but can correct them!. arXiv preprint arXiv:2311.08516.
>
> Stechly, K., Valmeekam, K., & Kambhampati, S. (2024). On the Self-Verification Limitations of Large Language Models on Reasoning and Planning Tasks. ArXiv, abs/2402.08115.
>
> Madaan, Aman, Niket Tandon, Prakhar Gupta, Skyler Hallinan, Luyu Gao, Sarah Wiegreffe, Uri Alon et al. "Self-refine: Iterative refinement with self-feedback." Advances in Neural Information Processing Systems 36 (2024).
> Ghosal, D., Shen, S., Majumder, N., Mihalcea, R., & Poria, S. (2022). CICERO: A Dataset for Contextualized Commonsense Inference in Dialogues. Annual Meeting of the Association for Computational Linguistics.

---

> > ### Author Response · Authors · 2024-08-12
> >
> > Thank you again for your valuable review. If you have any further questions or concerns, please let us know so we can address them before the end of the discussion phase. If you feel that our responses have addressed your original concerns, please consider updating your evaluation. We appreciate your time and effort in this process!

---

### Official Review · Reviewer_zQB8 · 2024-07-13

**Soundness:** 3
**Presentation:** 3
**Contribution:** 3
**Rating:** 6
**Confidence:** 4

**Summary:**

The paper presents AutoMix, a method designed to optimize the use of large language models (LLMs) by strategically routing queries to different models based on performance and cost considerations. AutoMix relies on a few-shot self-verification mechanism which involves the estimation of correctness of a smaller model's outputs. Thereafter, a router model determines whether to query a more powerful, larger model to generate the output or accepts the smaller model’s outputs. This router is POMDP-based, trained with a reward function that couples both accuracy and computational efficacy (penalizing use of larger model). Experiments conducted on five language models across five challenging datasets show that AutoMix significantly reduces computational costs while maintaining comparable performance levels, outperforming other computational efficient baselines.

**Strengths:**

The proposed approach is sound, intuitive and works even in the black-box setting.
Empirical results from five different dialogue and context-grounded reasoning tasks.show that the proposed approach (AutoMix) outperform existing computationally efficient LLM approaches.
The paper is well-written and easy to follow.

**Weaknesses:**

There is a need for a RL-trained (POMDP) router agent. The generalizability of this router to other tasks/data is unclear.

**Questions:**

Please discuss the weakness mentioned above.

**Limitations:**

Yes

---

> ### Author Rebuttal · Authors · 2024-08-07
>
> Thank you for taking the time to review the paper and provide feedback! We address your concern here:
>
> ### Generalizability of POMDP router to other tasks/data
>
> The POMDP router does not assume any specific task or dataset characteristics to work, as it relies only on self-verification confidence as input. Thus, our *POMDP router is generalizable to various datasets and tasks.*
>
> To further demonstrate the generalizability of our router, we evaluate out-of-domain generalization by training on one dataset and evaluating on others. Specifically, we train the router on one of the five datasets and evaluate it on the other four. We repeated the experiment for all five datasets and three SLMs. The results in different settings, as shown in the table below, indicate that our POMDP router consistently outperforms both FrugalGPT and HybridLLM:
>
> |          | Mistral-7b | LLama-13b | GPT-3.5  |
> |----------|------------|-----------|----------|
> | Automix  | **28.3**   | **31.5**  | **70.9** |
> | Frugal   | 12.5      | 0.0       | 14.3    |
> | Hybrid   | 2.4       | -2.8     | 7.6    |
>
>
> Table 1: Out-of-domain generalization across various SLMs for different methods. Scores are averaged across five datasets.
>
> The design of the POMDP router, along with these results, highlights the strong generalizability of our proposed router.

---

> > ### Author Response · Authors · 2024-08-12
> >
> > Thank you again for your valuable review. If you have any further questions or concerns, please let us know so we can address them before the end of the discussion phase. If you feel that our responses have addressed your original concerns, please consider updating your evaluation. We appreciate your time and effort in this process!

---

> > ### Comment · Reviewer_zQB8 · 2024-08-13
> > **Thank you**
> >
> > I thank the authors for their response. I acknowledge the comments and decided to keep my positive score with more confidence.

---

### Author Rebuttal · Authors · 2024-08-07

We thank all the reviewers for their valuable feedback! We are encouraged that they find our "approach sound and intuitive, our paper as easy to read and follow" (Reviewer zQB8), that "our approach to self-verification and POMDP-based routing is novel, our methodology is sound, and recognizing the importance of our proposed IBC metric in cost-efficiency analysis" (Reviewer E2SA), and that our paper "addresses an intriguing problem, and the proposed solution is technically sound and well-described" (Reviewer NEfd), and our "integration of POMDP is interesting and well-motivated" (Reviewer RAHf), and "our proposed approach is smart and the paper being very well-written and easy to understand" (Reviewer eJtm).

We have addressed reviewers' comments individually and look forward to further comments within the remainder of the discussion period.

As part of the rebuttal, we have also included certain additional analyses and experiments enumerated below:

1. We expand our analysis of the "Effect of Cost Ratio on AutoMix" (Section 5.4) to show how different cost ratios affect cost-performance curves in addition to the IBC metric.
2. To demonstrate the generalizability of our method to other domains, we evaluate our method on an additional commonsense reasoning dataset: CICERO, demonstrating significantly higher results than baselines.
3. We demonstrate the out-of-distribution generalization capability of our proposed POMDP router, while baselines performed very poorly.
4. We extend the analysis in Section 5.5 to show the effectiveness of POMDP over other routers and baselines on additional datasets and models.
5. There have been concerns on generalization ability of method. While the details are addressed in each reviewer comment, we reiterate that Automix was evaluated on 5 diverse datasets where performance of models range quite widely and with 3 different models. In addition, we have included another dataset in rebuttal period to further strengthen this. We would be happy to include any other dataset suggested by reviewer as applicable to our setting.


We enclose these additional experiments in the response PDF, along with individual reviewer responses.

---

### Decision · Program_Chairs · 2024-09-25

**Decision:**

Accept (poster)

**Comment:**

This paper introduces a new technique (AutoMix) to optimize LLM inference costs, by training a router that can decide to query larger models only when required (while sticking to smaller models when not needed to properly answer a question).

After the discussion period, where authors clarified several points and included additional results, all reviewers agree this is a practically useful approach with clear demonstrated benefits and no major weakness. I agree this is a meaningful contribution, which anyone looking to decrease their API cost usage could be interested in. As a result, I am recommending acceptance.

I would like to ask the authors to be sure to include all additional results and important clarifications in the camera-ready version.